# GLEE: A Framework and Benchmark for LLM Evaluation in Language-based Economics

## Abstract

Large Language Models (LLMs) show significant potential in economic and strategic interactions, where communication via natural language is often prevalent. This raises key questions: Do LLMs behave rationally? Can they mimic human behavior? Do they tend to reach an efficient and fair outcome? What is the role of natural language in the strategic interaction? How do characteristics of the economic environment influence these dynamics? These questions become crucial concerning the economic and societal implications of integrating LLM-based agents into real-world data-driven systems, such as online retail platforms and recommender systems. While the ML community has been exploring the potential of LLMs in such multi-agent setups, varying assumptions, design choices and evaluation criteria across studies make it difficult to draw robust and meaningful conclusions. To address this, we introduce a benchmark for standardizing research on two-player, sequential, language-based games. Inspired by the economic literature, we define three base families of games with consistent parameterization, degrees of freedom and economic measures to evaluate agents' performance (self-gain), as well as the game outcome (efficiency and fairness). We develop an open-source framework for interaction simulation and analysis, and utilize it to collect a dataset of LLM vs. LLM interactions across numerous game configurations and an additional dataset of human vs. LLM interactions. Through extensive experimentation, we demonstrate how our framework and dataset can be used to: (i) compare the behavior of LLM-based agents to human players in various economic contexts; (ii) evaluate agents in both individual and collective performance measures; and (iii) quantify the effect of the economic characteristics of the environments on the behavior of agents. We believe that our framework can contribute to the growing intersection of LLMs, ML, and economics, and we encourage researchers to explore it further and build on its foundation.

## 1 Introduction

Recent research has increasingly focused on the capabilities of Large Language Models (LLMs) in decision-making tasks, revealing their potential to operate as autonomous agents in various economic environments, that typically require involving complex strategic thinking (Horton 2023; Wang et al. 2023; Zhu et al. 2024; Li et al. 2024). Applications of LLM-based agents in such environments include Chess-playing (Feng et al. 2024), task-oriented dialogue handling (Ulmer et al. 2024), financial advisement (Lakkaraju et al. 2023) and beyond. The promising capabilities of LLMs in strategic decision-making call for the study of these agents' behavior through the lens of game theory, e.g. whether and when LLM-based agents naturally converge to Nash-equilibrium (Guo et al. 2024), and how well they learn to cooperate in repeated interactions (Akata et al. 2023). The rise of LLM agents also calls for a clear benchmark for assessing the extent to which such agent behaves rationally, which is addressed by a recent work of Raman et al. (2024).

An important property of many real-world economic environments is that communication between agents typically occurs through *natural language*. While economic modeling usually abstracts away these nuances for the sake of keeping the model simple and tractable, in practical scenarios the fact that natural language is involved may significantly affect the interaction outcome. For instance, in bargaining between two parties, the same offer can be interpreted very differently depending on how it is framed. Consider the following two offers for a business partnership: **(a)** *"We propose a*

*40-60 split as we believe your expertise will drive the majority of the success.*" **(b)** *"We propose a 40-60 split because we've already invested significant resources and need a lower share to maintain balance.*" Both propose the same numerical terms, but the framing in each case may lead to different reactions based on how the rationale is presented. A similar rationale applies to a broader range of economic scenarios. To illustrate this further, let us consider the following representative use cases:

- **Bargaining.** Alice and Bob co-own a startup valued at a million dollar, and must decide how to divide the proceeds from its sale. To reach an agreement, they take turns proposing how to split the shared value, with each delay in reaching a decision reducing the overall value for both parties. The interaction often involves free text in the proposal exchange.

- **Negotiation.** Alice aims to sell a product to a potential buyer, Bob. They negotiate by taking turns proposing prices, with Alice deciding whether to accept Bob's offer and sell, and Bob deciding whether to accept Alice's price and buy, continuing this process until they reach an agreement or the negotiation concludes. The negotiation often involves natural language, such as descriptions of the product, which can influence the outcome.[1]

- **Persuasion.** Alice is a seller trying to sell a product to Bob at an exogenously set fixed price. Alice knows the true quality of the product, while Bob only has a rough expectation of the product's quality. Alice sends a message to persuade Bob to buy the product, aiming to convince him of its value, regardless of its true quality. Bob, however, benefits only if the product is genuinely high quality. The interaction then repeats for multiple rounds, where product qualities are realized independently at different rounds. In such a scenario, Alice balances between her maximizing one-shot gain and building positive reputation, and linguistic communication can play a significant role.

These three types of interactions are inspired by influential models in the economics literature. They are also broad and flexible enough to capture a wide variety of real-world applications. The bargaining model, inspired by the seminal work of Rubinstein (1982), forms the basis for understanding how parties negotiate the division of shared resources, applicable in scenarios such as business mergers, partnership dissolution, and legal settlements. The negotiation game reflects celebrated models of bilateral trade (Myerson & Satterthwaite 1983; McAfee 1992) where buyers and sellers negotiate over the prices of an indivisible good, which often arises in various applications, including real estate transactions, corporate acquisitions, and e-commerce platforms. Lastly, the persuasion model draws on classical models of information asymmetry and strategic communication (Akerlof 1978; Crawford & Sobel 1982; Farrell & Rabin 1996), which play a pivotal role in advertising, marketing, political campaigning, and recommendation systems.

A key feature of all these games is that they are sequential, meaning players take turns acting rather than acting simultaneously. This makes communication, and in particular language-based communication, a crucial element of the interaction. In sequential games, communication is typically direct, with players able to attach messages to their actions (i.e., in the bargaining example, the textual message is attached to the proposal). In contrast, in simultaneous games, communication tends to be indirect, as players act independently and typically learn to cooperate by observing and reacting to past actions rather than through direct message exchange. A prime example of a simultaneous, language-based economic interaction is the competition among web publishers in search engines. Publishers simultaneously create content (usually in the form of textual documents) for their websites, and then gradually learn and adapt based on outcomes, such as their relative search engine rankings and exposure. However, they usually lack the opportunity for real-time communication or coordination during this process. These simultaneous, language-based interactions are well-studied within the information retrieval community, and the induced publishers' game is indeed modeled as a simultaneous game.[2] Our work is complementary, focusing on sequential settings where direct, language-based communication is prevalent.

---

[1]Notice that in the economic literature, the terms "bargaining" and "negotiation" are often used interchangeably to describe both the division of a divisible good and the process of price negotiation. In this paper, we use the terminology "bargaining" to describe the former and "negotiation" for the latter, for ease of exposition. We refer to Appendix A for further discussion on the differences between the two games.

[2]See e.g. Raifer et al. (2017); Ben-Porat et al. (2019); Kurland & Tennenholtz (2022); Hron et al. (2022); Madmon et al. (2023; 2024); Nachimovsky et al. (2024). In these works, SEO competitions are modeled as simultaneous games.

The AI and NLP communities have recognized the potential of integrating natural language into stylized economic models of bargaining, negotiation and persuasion. The also recognize the importance of studying the capabilities and limitations of LLM-based agents within such frameworks. A growing body of research is dedicated to evaluating and optimizing LLM-based agents in various bargaining and negotiation scenarios (Abdelnabi et al. 2023; Deng et al. 2024; Xia et al. 2024; Bianchi et al. 2024; Noh & Chang 2024). In the realm of persuasion games, recent studies have introduced language-based frameworks that explore the design of optimal information transmission policies (Raifer et al. 2022) and develop methods for generating data to predict human players' behavior (Apel et al. 2022; Shapira et al. 2023; 2024).

Although each of these economic setups has been studied in the context of LLM-based agents, the approaches have varied widely across different studies. Each paper often adopts distinct modeling and implementation assumptions, poses unique research questions, and applies diverse evaluation criteria, making it challenging to compare results and draw broader conclusions about the capabilities and limitations of LLM-based agents in economic environments. To enhance the real-world reliability of LLM-based agents, it is essential to establish a clear benchmark that provides a standardized framework for modeling these economic interactions. This benchmark would ensure comparability across studies and enable generalization of findings, facilitating a deeper understanding of how various factors influence interaction outcomes and leading to more robust and reliable conclusions about the performance of LLM-based agents in real-life economic situations.

## 1.1 OUR CONTRIBUTION

We introduce GLEE, a unified framework for **G**ames in **L**anguage-based **E**conomic **E**nvironments,[3] focusing on the case of two-player games.[4] Our framework relies on a clear and comprehensive parameterization of the space of all bargaining, negotiation and persuasion games (as described above), and defines degree of freedom and evaluation metrics that are consistent across economic contexts. The generality of the space of games spanned by our parameterization follows from the richness of the degrees of freedom considered, which include the game horizon (number of rounds within each game), information structure (whether agents know each others' preferences or not), and communication form (whether agents communicate via free language of structured messages).

Through this framework, we have collected a dataset of LLM vs. LLM games, containing 7.15M decisions made by LLM-based agents in over 954K games, using four different LLMs. Our framework allows for running controlled experiments across various degrees of freedom, language models, and game scenarios, making the dataset valuable for comparing and evaluating LLM performance in different economic setups. It also facilitates deeper insights into how characteristics of the economic environment influence agent behavior and ultimately shape the outcomes of interactions. Additionally, we design and develop an interactive interface we run experiments in which human players play against LLM-based agents in those considered games, for a variety of game parameters, and collect a human vs. LLM interaction dataset. We utilize this human-generated dataset to introduce a benchmark for several human choice prediction tasks, in which ML algorithms can be used to accurately predict the behavior of human players in a wide range of economic contexts.

To recap, the contribution of this paper is fourfold. First, we introduce a clear parametrization of a large, general, and representative set of sequential, two-player, language-based games. This parametrization is inspired by both the relevant economic literature, as well as advances in the LLM agents literature. Importantly, our parameterization can be naturally extended beyond the two-player case. Second, we release open-source code for GLEE, a unified framework that enables controlled experiments across the representative families of games considered. Notably, our framework is flexible and facilitates the collection of additional interactions with novel LLMs, game parameters and evaluation metrics, which is crucial given the fast advancements in the field. Third, we utilize GLEE to collect a dataset of LLM vs. LLM and human vs. LLM interactions in a wide range of economic environments, building on the foundations of our comprehensive parametrization. Finally, we

---

[3]Code and data are available in `https://github.com/gleeframework/GLEE`.

[4]The two-player case is particularly important since the main shift occurs when moving from single-agent decision-making to two-agent interactions, in which strategic behavior and communication play crucial roles. Two-player scenarios are also widely observed in real life, and the phenomena emerging from these interactions tend to be more intuitive and easier to analyze, while still capturing the core behaviors that arise in more complex multi-agent environments.

demonstrate how the dataset can be analyzed to draw meaningful conclusions and economic insights regarding LLM-based players and human players in language-based economic environments.

## 2 GAME FAMILIES AND PARAMETRIZATION

In this section, we formally define the three game families discussed in the introduction: bargaining, negotiation and persuasion.[5]

In our modeling, we focus on three main degrees of freedom that are critical for understanding the dynamics of LLM-based agents in economic interactions: *game horizon*, *information structure*, and *communication form*. The *game horizon* refers to the number of time periods during which the game is played and whether the length of the horizon is known or unknown to the agents.[7] This factor influences the strategies agents adopt, particularly in terms of long-term planning and anticipation of future moves. The *information structure* determines whether agents are aware of each other's preferences, impacting their ability to predict and respond to the actions of others. Lastly, the *communication form* specifies whether communication between agents occurs through free language or structured, concise messages, which affects the richness and clarity of the exchanges.

### 2.1 BARGAINING GAMES

The first family of games is inspired by the celebrated bargaining model of Rubinstein (1982). The model encompasses a class of bargaining games where two players, Alice and Bob, alternate offers over a time horizon $T$ (usually, $T = \infty$) to divide a fixed sum of money $M$ between them.[8] Importantly, in these games delays are costly, a concept captured by *discount factors* $\delta_A, \delta_B$ assigned to each player, reflecting the decreasing value of future payoffs as time progresses. Formally:

At each odd stage $t$, Alice offers a division $(p, 1 - p)$ for some $p \in [0, 1]$. Bob decides whether to accept or reject. If Bob accepts, the (Alice, Bob) utility vector is given by $M(\delta_A^{t-1} p, \delta_B^{t-1}(1 - p))$. At each even stage $t$, Bob offers a division $(q, 1 - q)$ for some $q \in [0, 1]$. Alice decides whether to accept or reject. If Alice accepts, the game terminates, and the (Alice, Bob) utility vector is given by $M(\delta_A^{t-1} q, \delta_B^{t-1}(1 - q))$. If no agreement is reached at any stage, the utilities are defined to be $(0, 0)$. In the standard version of the game, the time horizon is infinite and the discount factors are common knowledge (i.e., Alice and Bob know both $\delta_A$ and $\delta_B$).

In our experiments, we simulate a wide range of such bargaining games, differing in the following degrees of freedom: **(i)** whether or not the players know their opponent's discount factor (complete vs. incomplete information); **(ii)** whether or not the players know when the game terminates (finite vs. infinite); **(iii)** whether or not players communication involve natural language (structured vs. linguistic); **(iv)** the values of $\delta_A, \delta_B \in (0, 1)$ and $M$; **(v)** the value of $T$ in the finite horizon case.

An outcome of the game is a pair $(t_{ev}, p_{ev})$, where $t_{ev}$ is the stage index at which the game terminated, and $p_{ev}$ is the share of $M$ that Alice obtained (without considering the discount in the utilities). When the game terminates without reaching an agreement, we define $t_{ev} = \infty$, and the gain for both players is zero. The evaluation metrics used to assess the economic outcome are *efficiency* and *fairness*. Efficiency is now measured by the normalized sum of Alice's and Bob's discounted payoffs at the time of agreement: $\delta_A^{t_{ev}-1} p_{ev} + \delta_B^{t_{ev}-1}(1 - p_{ev})$ if $t_{ev} < \infty$, and 0 if

---

[5]In Appendix A we further discuss the related economic literature, and review some well-known theoretical results concerning these games. For each family, we provide a formal game-theoretic definition of the game (including the players, strategies and utilities), define degrees of freedom, and define the game outcome and evaluation metrics. While previous research has effectively tackled the challenge of evaluating the rationality of individual agents (Raman et al. 2024), our focus extends to evaluating the outcome reached by the strategic behavior of the agents, using the fundamental economic notions of *efficiency* and *fairness*. [6]

[7]In economic theory, "infinite horizon" typically refers to a large, unspecified duration. Accordingly, we use the term "infinite horizon" to describe cases where the horizon is both large and unknown to the agents.

[8]It is clear that under full rationality assumption, the amount of money does not play a role in the analysis. However, for human (or LLM) players, it is evident that the amount of money indeed matters (See Figures 16, 22, 28 in Appendix E).

$t_{ev} = \infty$. Fairness is defined as the distance between the actual division and the fairest division, $1 - 4 \cdot (p_{ev} - \frac{1}{2})^2$ if $t_{ev} < \infty$, and 1 if $t_{ev} = \infty$.[9]

## 2.2 NEGOTIATION GAMES

In the second family of games, referred to as negotiation games, a seller (Alice) and a buyer (Bob) are negotiating over the price of a product. At the outset, Alice owns a product she subjectively values at $V_A$. The subjective valuation of the potential buyer, Bob, is $V_B$. To capture the notion of valuation scale in negotiation games, we parameterize $V_i = M \cdot F_i$ for $i \in \{A, B\}$, where $F_i \in (0, 1)$ is a factor parameter $M$ is a scale parameter.

As in the case of bargaining games, Alice and Bob alternate offers: at each odd stage, Alice posts a price and Bob decides whether to buy the product or move on to the next stage. At each even stage, Bob is the one to post a price and Alice decides whether to sell at this price or reject and move on to the next stage. The game is played for $T$ stages, which again can be either finite or "infinite" (i.e., large and unknown to both players). Unlike the bargaining game, here we assume no discount factors on the utilities, hence whenever a price $p$ is accepted, the utilities for Alice and Bob are $p - V_A$ and $V_B - p$ respectively. If no trade is made, then the utilities are defined to be $(0, 0)$.

In this class of games, we consider the following degrees of freedom: **(i)** whether or not the players know their opponent's product valuation (complete vs. incomplete information); **(ii)** whether or not the players know when the game terminates (finite vs. infinite); **(iii)** whether or not players communication involve natural language (structured vs. linguistic); **(iv)** the values of $F_A, F_B \in (0, 1)$ and $M$; **(v)** the value of $T$ in the finite horizon case.

An outcome of a negotiation game is captured by $p_{ev}$, which is the price at which the product is sold when there is a trade, and defined to be $p_{ev} = \emptyset$ whenever no trade is made. We consider the following evaluation measures of the game outcome *fairness* and *efficiency*. When there is trade, fairness is measured by $1 - 4 \cdot \left( \frac{p_{ev} - p_f}{M} \right)^2$, where $p_f = \frac{V_A + V_B}{2}$ is the "fairest price".[10] When trade is not made, we define the fairness to be 1 (i.e., maximal fairness) to reflect that no-trade does not change the default allocation of the product. Efficiency is defined to be 1 in the following cases (and zero otherwise): **(a)** Alice values the product more than Bob, and does not sell it ($V_A \geq V_B$ and $p = \emptyset$); **(b)** Alice values the product less Bob, and sells the product at a price that is beneficial for both players ($V_A \leq p \leq V_B$); When averaged over a large number of simulated games for a certain game configuration, this measure estimated the probability of the event "an efficient trade occurs when it should occur".

## 2.3 PERSUASION GAMES

In a persuasion game, a seller (Alice) tried to persuade a buyer (Bob) to buy a product at a fixed price $\pi$. Alice privately knows the true product quality, which can be either high or low. Bob only knows that the prior probability of the product being of high quality is $p$. Alice gets a utility of 1 if Bob buys (regardless of the true product quality), and 0 otherwise.

Bob values a high-quality product at $v > \pi$ and a low-quality product at $u < \pi$. Therefore, the utility of Bob from buying a high-quality product is $v - \pi > 0$, and from buying a low-quality product is $u - \pi < 0$. For simplicity, we normalize the price to $\pi = 1$ and the value of a low-quality product to $u = 0$. In addition, we consider a currency scale parameter $M$ that serves as a multiplicative term of Bob's utility function. Overall Bob gets a utility of $M(v - 1)$ from buying a high-quality product, $-M$ from buying a low-quality product, and 0 from not buying the product.

The timing of a single round is as follows. First, Alice observes the product quality (which is realized to be high-quality w.p. $p$, independently of other rounds). Then, Alice sends a message to Bob. Lastly, Bob decides whether to buy or not to buy the product, and utilities are realized accordingly. The game then consists of $T$ such rounds. Alice's goal is to maximize her cumulative

---

[9]We consider the case of no trade as fair, since both players get the same utility. Obviously, this is also the least efficient outcome, which highlights the natural fairness-efficiency tradeoff.

[10]Notice that unlike all other metrics, the fairness metric in negotiation games is not normalized, as the price $p_{ev}$ is unbounded in principle. However, we observe that normalizing the difference $p_{ev} - p_f$ by the scale parameter $M$ results in a measure that is between 0 and 1 in 99.5% of the cases.

utility over time. We differentiate between two types of persuasion game setups: **(a) Long-living buyer.** The buyer is long-living, in the sense that he also aims to gain his cumulative utility over time. In this case, both players observe the entire interaction history; **(b) Myopic buyers.** Buyers are myopic, in the sense the buyer of stage $t$ only cares about the utility obtained at stage $t$. In this case, each buyer observes the statistics of all previous rounds (i.e., % of rounds in which Bob bought the product, and % of rounds in which Bob bought a low-quality product).[11]

We consider the following degrees of freedom in persuasion games: **(i)** whether or not Alice knows Bob's high-quality product valuation $v$ (complete vs. incomplete information); **(ii)** whether or not the players know when the game terminates (finite vs. infinite); **(iii)** whether or not Alice's messages involve natural language (structured vs. linguistic); **(iv)** the values of $v$, $p$ and $M$; **(v)** the value of $T$ in the finite horizon case; **(vi)** whether the game is with a long-living Bob or with myopic buyers.

An outcome of the game is a tuple $(n_{ev}, k_{ev}, r_{ev})$, where $n_{ev}$ is the number of rounds in which the product was of high-quality, $k_{ev}$ is the number of rounds in which the product was of high-quality *and* the buyer bought the product, and $r_{ev}$ is the number of rounds in which the product was of low quality *and* the buyer did not buy the product. We define *efficiency* to by the proportion of rounds in which the product was sold out of all rounds in which the product was of high quality (i.e., $\frac{k_{ev}}{n_{ev}}$), and *fairness* to be the proportion of rounds in which the product was not sold out of all rounds in which the product was of low quality (i.e., $\frac{r_{ev}}{T-n_{ev}}$).

## 3 DATA COLLECTION

In this section, we describe the data collection process. We developed a game management system to facilitate data collection from the games described in §2. The system is written in Python and is easy to use and customize, allowing future researchers to collect data seamlessly. New language models can be added easily, enabling them to play in any configurable setup. With a single command line, the system allows running either a single configuration or a set of configurations. Prompt samples are described in Appendix B.

### 3.1 LLM DATA COLLECTION

| | **Bargaining** | | **Negotiation** | | **Persuasion** |
|---|---|---|---|---|---|
| $\delta_A$ | 0.8, 0.9, 0.95, 1 | $F_A$ | 0.8, 1, 1.2, 1.5 | p | $\frac{1}{3}$, 0.5, 0.8 |
| $\delta_B$ | 0.8, 0.9, 0.95, 1 | $F_B$ | 0.8, 1, 1.2, 1.5 | v | 1.2, 1.25, 2, 3, 4 |
| $M$ | $10^2, 10^4, 10^6$ | $M$ | $10^2, 10^4, 10^6$ | M | $10^2, 10^4, 10^6$ |
| $T$ | $12, \infty$ | $T$ | $1, 10, \infty$ | $T$ | 20 |
| CI | True, False | CI | True, False | CI | True, False |
| MA | True, False | MA | True, False | Messages type | Binary, Textual |
| | | | | Buyer type | Long-living, Myopic |
| In total | 384 configurations | In total | 576 configurations | In total | 384 configurations |

Table 1: Parameters and their optional values used to define the 1,344 game configurations across bargaining, negotiation, and persuasion game families for data collection. $T = \infty$ means a very large value of $T$, which is unknown to the players. CI = Complete Information. MA = Textual messages allowed.

Since the game space defined by the parameters presented in §2 is infinite for each of the game families, it is clear that data cannot be collected from all possible games. Therefore, we attempted to cover the game space by selecting diverse values for each of the parameters defining the games, and we collected data from every possible combination generated by these parameters.

Table 1 shows the parameters defining the groups from which we collected data. In total, we collected data from 1,344 different configurations: 384 configurations of bargaining, 576 configurations of negotiation, and 384 configurations of persuasion games.

---

[11]That is, each buyer observes sufficient statistics from the entire history. This implementation detail is due to context length memory which is an inherent limitation of LLM agents, as well as to reducing cognitive load on human players.

| | human | Qwen-2 | Gem | Lam-3 | Lam-3.1 | Barg. | Nego. | Pers. | total |
|---|---|---|---|---|---|---|---|---|---|
| # Games | 3.4K | 309K | 429K | 342K | 332K | 282K | 539K | 133K | 954K |
| # Messages | 6.64K | 1.16M | 870K | 1.34M | 1.33M | 666K | 1.41M | 4.01M | 6.09M |
| # Words | 58.8K | 40.9M | 29.2M | 51.7M | 54.7M | 10.2M | 52M | 174M | 236M |
| # Decision | 7.43K | 1.42M | 1.11M | 1.58M | 1.56M | 1.01M | 2.15M | 4M | 7.15M |

Table 2: Statistics of data collected: by human players and each of the LLMs (left), for each game family (center), and in total (right). The number of games indicates the number of games in which the language model was involved as one or both players. The number of messages and decisions represents the number of decisions made and messages sent by the language model. Gem. = Gemini, Lam = Llama; Barg. = Bargening, Nego. = Negotiation, Pers. = Persuasion

For data collection, we used four language models: Qwen-2-7B (Yang et al. 2024), Llama-3-8B, Llama-3.1-8B (Meta 2024), and Gemini-1.5-Flash (Gemini Team 2024). Each of the 16 possible pairs (including a language model against itself, with attention to the order defining the roles of the players in the game) played at least 30 games from the 1,344 configurations. In total, 954K games were played, with full statistics appearing in Table 2. The complete data from all games, including the messages sent by the language models, the decisions made, and the prompts and responses of the language models that generated these messages, form one of the main contributions of this paper and are available in the paper's GitHub repository.

In addition to these models, we collected data for some of the configurations using Gemini-1.5-Pro (Gemini Team 2024) and Llama-3.1-405B (Meta 2024), two larger, more powerful, and expensive models compared to the ones used. These models played against each other and against some of the four presented models. Due to budget constraints, data collection with these models was not comprehensive, but the data from these games still contributes to the paper.

It was possible to alter the prompt that introduces the game, explore different variations, and allow the language models to play different personas. While this could impact the behavior of the language models, budget limitations would have required us to collect less data from each configuration. Therefore, we chose to keep the prompt fixed rather than further defining the language model's setup. However, the system allows easy prompt customization to support future research.

## 3.2 HUMAN DATA COLLECTION

One of the main objectives of collecting data from language games is to compare the behavior of LLMs with human behavior in economic and strategic situations. To facilitate this comparison, we developed an interface that allows human players to play all the language games that can be defined using GLEE. The interface transforms the various prompts presented to LLMs into user-friendly screens for human participants, displaying the prompt and requesting them to send messages and make decisions.[12] Through this interface, we enable human players to take on the role of one of the players while the other player is a pre-selected LLM. The interface is one of the contributions of this paper. Screenshots of the interface can be found in Appendix C.1.

The interface, developed using oTree (Chen et al. 2016), enables integration with Amazon's crowd-sourcing platform, mTurk,[13] through which we recruited 3,405 players who participated in various configurations against Gemini-1.5-flash. We chose Gemini-1.5-flash since it demonstrated strong performance compared to other LLMs and allowed comprehensive data collection due to its low usage cost. Since we aimed to collect multiple games from each configuration, we had to select a limited set of configurations for human participants to play. The process of selecting these sets is described in Appendix C.2. We collected human data from 195 different configurations: 78 of which were bargaining games, 60 of which were negotiation games, and 57 were persuasion games.

Each human player was allowed to play one game every 12 hours from each family of games. Human players were paid a base rate calculated at $6 per hour, plus an average bonus of $6 per

---

[12]Since human players are accustomed to being addressed by their first name (rather than as Alice or Bob), we asked them to enter their name at the beginning of the game and referred to them by their name throughout the game. The player's name was the only difference between the human player and the LLM-based player.

[13]https://www.mturk.com/.

hour. In total, we paid $2,245 to all players for their participation in the games. Bonuses were dependent on the configuration the human players played and their success in the game. The average bonus was known to players at the start of the game, aiming to encourage serious gameplay. To ensure players remained focused and made thoughtful decisions, we conducted two attention checks during the experiment, detailed in Appendix C.3. Players who failed in one of the attention checks were excluded from the dataset. To reflect the real-world significance of the magnitude of product prices (the parameter M) in each family of games, we defined the bonus as dependent on M: in configurations where $M = 10^2$, the average bonus was \$3 per hour; where $M = 10^4$, the average bonus was \$6 per hour; and where $M = 10^6$, the average bonus was \$9 per hour.

## 4 EXPLORATORY DATA ANALYSIS

In this section, we first present a method to ensure an adequate comparison between models that played different configurations. Then, we address two key research questions: **Q1:** How did the language models perform in each game family in terms of self-gain, efficiency, and fairness? **Q2:** How did the humans perform compared to the language models? **Q3:** How do the parameters defining the games influence the various metrics?

**Adequate Comparison Between Models** For the four language models introduced in §3, we collected data covering the entire set of the configurations defined in Table 1 for every possible pair of language models. This enables seamless comparison of their performance across all defined metrics. Henceforth, this dataset will be referred to as the baseline set, and the models that generated it will be referred to as the baseline models.

When evaluating data generated by a new model or by humans, it is often not feasible (due to cost considerations) to have them play all games against all existing language models.[14] To evaluate model performance on a large variety of language games, we can collect data from a subset of configurations, calculate the metrics for those, and use the results (along with the metrics from the baseline set) to train a predictive model that estimates the metrics for configurations for which data was not collected. This method allows for an adequate comparison between models that played different configuration sets.

For the calculation of the metrics, we train a simple regression model for each metric. The model receives as input the values defining the configuration as binary features (e.g., "Is $M = 100$?") and binary features describing the interaction between the language models and other parameter values (e.g., "Did Alice played by Gemini-1.5-Flash and $M = 100$?" and "Did Alice played by Gemini-1.5-Flash, Bob played by Llama-3.1-8B, and $M = 100$?"). To prevent the model from learning relationships between parameter values directly from configuration settings, we used binary features even for parameters that contain numerical values. We trained the models on the data from all games and predicted the values for all configurations: both those for which data was collected and those for which it was not. Therefore, to maintain consistency, all the metrics are calculated by averaging the predicted values over all game configurations. This approach allows us to reasonably estimate the metric values expected for configurations where no human data was collected. The exact formulas defining each regression model are presented in Appendix D.

To assess the quality of the models, we examined the adjusted R-squared values of the regression models. In 10 out of 12 trained models, we obtained adjusted R-squared values higher than 0.38. Table 3 shows the adjusted R-squared values of the models. The table shows that, in general, persuasion games are the easiest to predict, and we hypothesize that this is because their outcomes consist of a collection of multiple decisions. Negotiation games are the most difficult to predict, possibly because, unlike the other games, players could exit the game at any given moment. It seems that fairness is the most challenging metric to fit with a regression model, with negotiation games showing an especially low adjusted R-squared.

**Model and Human Performance (Q1 and Q2)** Table 4 presents, for each language model, the metric values for each role in each game family. The presented values are the predicted average values over all configurations for which data was collected, as described in §3. For each role,

---

[14]Collecting data from humans playing 30 games in each possible configuration would cost approximately \$187K. Collecting similar data from Gemini-1.5-Pro would cost around \$10K.

|  | Alice gain | Bob gain | Efficiency | Fairness |
|---|---|---|---|---|
| Bargaining | 0.54 | 0.47 | 0.57 | 0.28 |
| Negotiation | 0.48 | 0.58 | 0.38 | 0.09 |
| Persuasion | 0.63 | 0.89 | 0.58 | 0.41 |

Table 3: Adjusted R-square values of the regression models used for metric calculations and analyzing the impact of specific parameters on metrics.

we present the self-gain achieved by the model, as well as the two metrics introduced in §2: the efficiency and fairness values observed in games involving the model. Note that while the self-gain metric measures the player's performance in the game, the efficiency and fairness metrics describe the economic quality of the interaction outcome. This table allows us to compare the performance of different language models across game families.

The table allows us to compare the performance of humans to that of language models across different game families. Humans outperformed the language models in bargaining games when playing as Alice, but when playing as Bob, they achieved the worst performance. In negotiation games, humans performed poorly, with a negative average self gain, in contrast to the language models, which managed to achieve gains in both roles (except for Qwen-2 in the role of Alice). Human performance in persuasion games was relatively strong, second only to that of Llama-3.1.

Importantly, metric values presented in Table 4 are calculated by averaging over all possible game configurations in our dataset. These averages are therefore highly sensitive to the particular configurations, and the configuration distributions, in our dataset. To tailor the benchmark to specific applications, we recommend re-defining the parameter space and their distributions according to the economic context.[15] In addition, notice that the fact that metrics are averaged over all configurations

| Family | Rule | Metric | Human | Qwen-2 | Lam-3 | Lam-3.1 | Gem-F |
|---|---|---|---|---|---|---|---|
| Bargaining | Alice | Self Gain | **0.61** | 0.49 | 0.43 | 0.46 | 0.55 |
|  |  | Efficiency | **0.89** | 0.88 | 0.86 | 0.86 | 0.88 |
|  |  | Fairness | 0.71 | **0.87** | **0.87** | 0.86 | 0.76 |
|  | Bob | Self Gain | 0.27 | 0.34 | 0.38 | 0.42 | **0.43** |
|  |  | Efficiency | 0.94 | **0.97** | 0.79 | 0.78 | 0.88 |
|  |  | Fairness | 0.69 | 0.81 | 0.86 | 0.80 | **0.91** |
| Negotiation | Alice | Self Gain | -0.24 | -0.02 | 0.03 | 0.05 | **0.07** |
|  |  | Efficiency | 0.65 | 0.68 | **0.75** | 0.73 | **0.75** |
|  |  | Fairness | 0.39 | 0.82 | 0.89 | **0.91** | 0.88 |
|  | Bob | Self Gain | -0.05 | 0.08 | 0.05 | 0.07 | **0.12** |
|  |  | Efficiency | 0.67 | 0.73 | 0.67 | 0.69 | **0.81** |
|  |  | Fairness | 0.58 | 0.81 | 0.83 | **0.84** | 0.82 |
| Persuasion | Alice | Self Gain | 0.55 | 0.60 | 0.57 | **0.61** | 0.58 |
|  |  | Efficiency | 0.55 | **0.78** | 0.71 | 0.76 | 0.65 |
|  |  | Fairness | 0.41 | **0.63** | 0.61 | 0.60 | 0.53 |
|  | Bob | Self Gain | 0.68 | 0.45 | 0.53 | **0.75** | 0.41 |
|  |  | Efficiency | **0.93** | 0.84 | 0.68 | 0.35 | 0.65 |
|  |  | Fairness | 0.21 | 0.59 | 0.60 | 0.61 | **0.77** |

Table 4: Average predicted self-gain, efficiency, and fairness metrics for each language model across different roles and game families, comparing baseline models and human players. In bold: The highest value achieved in the metric.

---

[15]For instance, one could ask whether agents' performance significantly differs in economic environments where inflation is high (translating into lower discount factors, in bargaining games). To evaluate such a scenario, one can simulate games in which discount factor distribution fits these conditions, and re-evaluate agents' performance with respect to the new distribution of configurations.

may hide the effect of particular game parameters on the game outcome or agents' self-gain. We next demonstrate how a more delicate analysis of the dataset enables to isolation of the effect of certain game parameters.

**Influence of Game Parameters (Q3)**   To evaluate the effect of a specific parameter on a metric, we use the same regression model. To create a graph depicting the effect of a parameter X on a metric Y achieved by a language model Z playing as Alice, we extract from the regression model the effects of X values, Z values, and their interaction on Y, summing the relevant values. The resulting depiction shows the impact of parameter changes relative to a baseline configuration defined by one of X's values. Appendix E presents graphs describing the impact of each parameter on player gains, fairness, and efficiency in games, depending on the language model that played the games. The graphs include confidence intervals at a 95% confidence level.

Figure 1 serve as example from which conclusions can be drawn about the impact of the parameters. Figure 1 (Right) shows that full information reduces efficiency for all models in bargaining games (and also in negotiation games, as shown by Figure 24 in Appendix E). While this may be counter-intuitive at first, it might be the case that complete information prevents trade when efficient trade is possible.[16] Figure 1 (Left) shows that sending textual messages increases efficiency in negotiation games (a trend that also appears in bargaining games, as shown by Figure 19 in Appendix E). In persuasion games, sending textual messages improves Alice's gains for all models (as can be seen in Figure 31 in Appendix E), a result observed by Apel et al. (2022) in language-based games involving humans.

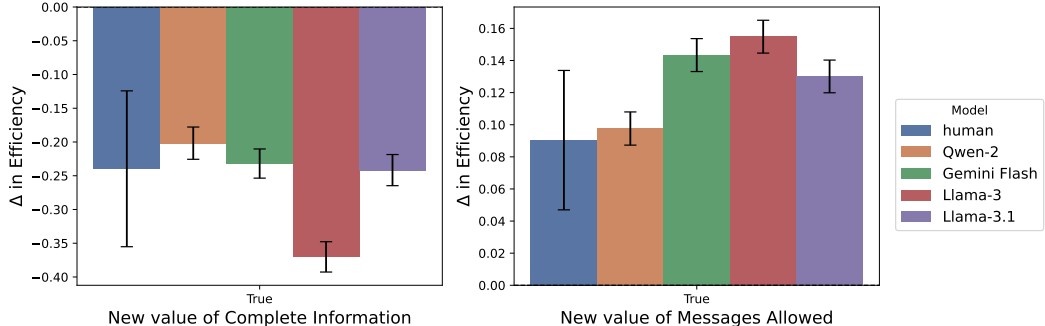

Figure 1: Right: the negative effect that full information has on efficiency in the bargaining game, depending on the player model. Left: the positive effect that sending textual messages has in negotiation games, depending on the player model.

## 5 CONCLUSION

We present GLEE, a framework for evaluating the behavior of Large Language Models (LLMs) in language-based economic games. The goal of GLEE is to provide a comparative tool for assessing the performance of LLMs in various economic scenarios and enable their comparison to human players. We defined the game space within three main families of games: bargaining, negotiation, and persuasion, and introduced metrics to measure player performance. We developed a framework that allows for large-scale data collection from games between diverse LLMs and created an interface that facilitates the collection of data from games involving human players. Through this interface, we gathered data from 954K games between LLMs and from 3,405 games involving human players. The data is available for future research, which could advance the field of machine learning in language-based economic games, such as for predicting human decisions using artificial data and building more successful and human-like agents based on the metrics we define in GLEE.

---

[16]Suppose, for example, that Alice values the product at 80$ and Bob values the product at 100$. If Alice sets a price of 99$, and Bob knows Alice's valuation, then Bob might reject the offer even though it is beneficial to him, since he feels that Alice should have made a more decent offer that takes into consideration Bob's satisfaction as well.

ETHICS STATEMENT

This paper aims to provide a platform for experimenting with agents in language-based economic environments. Naturally, this line of research may have various societal and ethical implications, as we now discuss.

First, studying the economic aspects of LLM-based agents has the potential to enhance the ability of agent designers to control and optimize the behavior of these agents. This capability can be utilized for a variety of purposes, ranging from encouraging self-interested behavior at the expense of other participants in the environment (e.g., for maximizing revenue in competitive settings) to promoting efficient trade and fair behavior, or any combination of these sometimes non-aligned objectives. As this research increases the power of LLM-based agents in economic environments, it is essential to emphasize that with great power comes great responsibility. We call for the responsible and ethical use of these emerging capabilities to ensure they are leveraged for socially beneficial purposes rather than exploitative ones.

Furthermore, our framework demonstrates the capability of collecting data from human players to understand the differences and similarities between LLMs and humans in economic environments. While this line of research has the potential for a strong scientific contribution, particularly in the field of behavioral economics, it also raises several ethical considerations. The collection of human data must be conducted with careful regulation and adherence to clear ethical guidelines, such as those established in venues like ICLR. The authors declare that there are no violations of the ICLR ethics code in our study. The entire process of data collection from human participants is elaborated in §3.2.

In addition to the challenges associated with the collection of human data, enhancing our understanding of LLMs from the lens of human behavior carries inherent risks. For instance, this research could enhance our ability to design LLM agents that are difficult to distinguish from real humans. Such capabilities could be misused for malicious purposes, including deception or manipulation. While the answer to whether these capabilities could be used for harmful causes is likely yes, we believe that the benefits of pursuing this line of research outweigh the risks when balanced with proper regulations. We advocate for pushing research forward while ensuring that any new technologies are accompanied by safeguards to prevent harmful usage, particularly when human-like LLMs are involved.

Our proposed framework can make these research directions more accessible to researchers from the ML community and beyond, thereby encouraging a broader understanding of LLM behavior in economic contexts. However, as such accessibility increases, it is crucial to maintain ethical oversight and foster an open dialogue on potential misuse. We encourage researchers to use our framework with full transparency and careful attention to potential misuse and negative consequences.

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

## A  GAME FAMILIES THROUGH THE LENS OF ECONOMIC LITERATURE

In this section, we discuss the economic literature related to the three game families considered in this paper, and review some known theoretical results.

**Bargining**  As mentioned in §2, the standard bargaining model of Rubinstein (1982) consists of two players, Alice and Bob, engaging in alternating offers for a finite horizon ($T = \infty$) with commonly known discount factors $\delta_A, \delta_B$. Our parameterization considers several additional degrees of freedom, such as finite vs. infinite time horizon, complete vs. incomplete information, and free language messages vs. structured and concise messages. In the standard model, Rubinstein (1982) showed that in the unique subgame-perfect equilibrium an agreement is reached in the first stage (i.e., utilities are not discounted), and the share of Alice is given by $Mp^*$, where $p^* = \frac{1-\delta_B}{1-\delta_A\delta_B}$.[17] The case of finite horizon can be solved using backward induction, and typically results in a different outcome compared to the infinite case. As $T$ grows, the equilibrium outcome approaches the one of the infinite case. Extensions to incomplete information regarding the opponent's discount factor are typically more challenging, and some of them are studied in the literature, e.g. Rubinstein (1985).

**Negotiation**  In a negotiation game, Alice and Bob negotiate over the price of an indivisible good. The negotiation game differs from the bargaining games in several key aspects:

1. Negotiation involves an indivisible product (e.g., Alice's product), while bargaining focuses on dividing a divisible resource, such as money.

2. In negotiation, Alice and Bob may have different subjective valuations of the product, whereas in bargaining, both parties value the divisible resource similarly.

3. Negotiation has no discounting, so the utility remains constant over time. In bargaining, delays reduce the total value, encouraging faster agreement.

If the seller is uncertain regarding the buyer's valuation but has a prior belief distribution, a classical result by Harris & Raviv (1981) and Riley & Zeckhauser (1983) states that it is always optimal to sell the product at a fixed price. In contrast, if the seller does not have a prior belief over the buyer's valuation, and instead aims to minimize regret, then an optional pricing policy will be to randomly choose a price from a carefully chosen distribution (Bergemann & Schlag, 2011).

**Persuasion**  Our persuasion game follows the structure of a cheap talk game (Crawford & Sobel, 1982; Farrell & Rabin, 1996), where the sender (Alice) cannot commit to a signaling policy in advance, unlike in Bayesian persuasion models (Kamenica & Gentzkow, 2011). Under the particular payoff structure considered in our persuasion game, it is well-known that the cheap-talk game only admits a *babbling equilibrium*, i.e., an equilibrium in which all information is kept hidden (this is due to the strong misalignment of interests between the two players). In contrast, if the seller can *commit* to a signaling policy at the outset, as in Bayesian persuasion, then there exists a subgame-perfect equilibrium in which the seller commits to the following policy: When the product is of high quality, the seller recommends buying the product with probability 1. When the product is of low quality, then the seller recommends with probability $q = \min\{\frac{p}{1-p}(v-1), 1\}$. This policy is also incentive-compatible, in the sense that the buyer always buys the product when the seller recommends buying.[18] While the long-living buyer case is well-studied in the economic literature, such games often admit multiple equilibria, which makes the games difficult to analyze and predict (Kim, 1996; Aumann & Hart, 2003). As for the case of myopic buyers, Best & Quigley (2024) draws a connection between the repeated cheap talk game and the case of one-shot Bayesian persuasion, leading to an elegant analytical solution of the repeated game. Intuitively, the repetitive nature of the game induces a reputation effect, which plays a similar role to the commitment power in standard one-shot Bayesian persuasion.

---

[17]A subgame-perfect equilibrium is a strategy profile in which every player responds optimally in every hypothetical subgame of the game, including off-path scenarios that are not reached in practice. This solution concept can be seen as capturing a higher level of rationality compared to the alternative of Nash equilibrium.

[18]In fact, the probability of lying $q$ is determined such that the buyer is indifferent upon receiving a recommendation, taking into account his belief updating, which relies on using Bayes's law and knowing the seller's committed policy.

# B  DATA COLLECTION SYSTEM

In this appendix, we present an example of the conversation structures with the language models used for data collection in *GLEE*. The example is from a bargaining game, but the system's operating principle is the same in negotiation and persuasion games as well.

The prompts are divided into two types: (1) instructions and guidelines sent to the language model as a system message, and (2) game management messages sent to the language model as a user message. The language model performed its action, which was documented in the prompt as an assistant message. The system message begins with precise game instructions, which are constructed at the start of the game using the parameters that define the configuration. This is followed by a guideline specifying the format in which the LLM is required to respond. Each time the language model is required to make a decision (of any type), the language model receives a message containing all the information that has accumulated since its previous turn. This information appears as a system message and sometimes includes text written by the language model that played the second player.

Throughout the appendix, the system prompt (or parts of it) that Alice received are highlighted in red. Game management messages Alice received are shown in purple, and her response is shown in magenta. The system prompt that Bob received is highlighted in blue. Game management messages Bob received are shown in teal, and his response is shown in cyan.

---

Instructions to Alice:

Let's play a game. In this game, there are two players. You are playing as Alice. You and Bob need to divide 1,000$ between you. In each round of the game, one of the players will make a proposal for division and can describe their proposal in a few words. The other player will decide whether to accept the offer and finish the game, or to proceed to the next round, in which they will be the one making the proposal. We will relay messages between you and Bob. Beware of inflation! With each passing round, the money is worth 0% less for you. For Bob, the money is worth 10% less. You have 10 rounds to divide the money, or both of you will get nothing!

Instructions to Bob:

Let's play a game. In this game, there are two players. You are playing as Bob. You and Alice need to divide 1,000$ between you. In each round of the game, one of the players will make a proposal for division and can describe their proposal in a few words. The other player will decide whether to accept the offer and finish the game, or to proceed to the next round, in which they will be the one making the proposal. We will relay messages between you and Alice. Beware of inflation! With each passing round, the money is worth 10% less for you. For Alice, the money is worth 0% less. You have 10 rounds to divide the money, or both of you will get nothing!

---

Alice turn.

Alice got this message from the system:

'Round 1
Send your offer to divide $1,000 and a message to Bob.'

Alice guideline:

Send your offer to divide 1,000$ and the message you attached in the JSON format:
{"alice_gain": The part that you will receive in your offer,
"bob_gain": The part that Bob will receive in your offer,
"message": The message you pass to Bob}

Alice's response:

```json {"alice_gain": 900, "bob_gain": 100,
"message": "Let's start fair. I'll take the bigger share, but you get something too."} ```

---

Bob turn.

Bob got this message from the system:

Round 1
Alice's offer:
# Alice's message: Let's start fair. I'll take the bigger share, but you get something too.
# Bob gain: 100
# Alice gain: 900
Do you accept this offer?

Bob guideline:

Answer with {"decision": "accept"} or {"decision": "reject"}

Bob's response:

'''json {"decision": "reject"} '''

---

Bob turn.

Bob got this message from the system:

You have chosen to reject Alice's offer from round 1.
Round 2
Send your offer to divide $1,000 and a message to Alice.

Bob guideline:

Send your offer to divide 1,000$ and the message you attached in the JSON format:
{"bob_gain": The part that you will receive in your offer,
"alice_gain": The part that Alice will receive in your offer,
"message": The message you pass to Alice}

Bob's response:

'''json {"bob_gain": 500,
"alice_gain": 500,
"message": "Let's split it evenly. It's the fairest way to start."} '''

---

Alice turn.

Alice got this message from the system:

Bob rejected your offer from round 1.

Round 2
Due to inflation, the money Bob gains is worth 10% less than in the first round. The money you gains is worth the same as in the first round.
Bob's offer:
# Bob's message: Let's split it evenly. It's the fairest way to start.
# Alice gain: 500
# Bob gain: 500
Do you accept this offer?

Alice guideline:

Answer with {"decision": "accept"} or {"decision": "reject"}

Alice's response:

'''json {"decision": "accept"} '''

The game is over.

## C    DATA COLLECTION FROM HUMAN PLAYERS

This appendix provides additional information on the human data collection interface described in §3.2, which facilitates data collection from `GLEE` games played between humans and LLMs.

### C.1    SCREENSHOTS OF THE DATA COLLECTION INTERFACE

**General application structure**    The structure of the application and the games consists of fixed parts and parts that vary between different game families. Each game starts with a screen where the player enters their name, followed by an instruction screen (Figure 2 in bargaining, Figure 6 in negotiation, and Figure 10 in persuasion). The instructions themselves can change depending on the type of game and the parameters of the game. On the instruction screen, there is a hidden prompt to enter a code word in the text box below, which is designed to filter out unfocused participants. If the player fails this test, they are taken to a screen that informs them of their failure. Otherwise, the game itself begins. In each round, both the human player and the LLM player perform an action, with the order depending on the game and configuration. An action could involve sending an offer to the other player (figure 3 in bargaining games, Figure 7 in negotiation games and Figure 11 in Persuasion games) or responding to the other player's offer (for instance, Figure 4 in bargaining games, Figure 8 in negotiation games and Figure 12 in persuasion games). After both players have completed their actions, the human player is taken to a response screen (Figure 5), where he sees the decision of the LLM player. Afterward, if the game is still not over, the human player continues for another round. Once the game is finished, the player is taken to a quiz screen where they must answer a question related to the technical details of the game (Figure 9). If the human answer correctly, they are directed to the final screen (Figure 13), where they receive a code to enter on the mTurk website. If the player fails the final quiz, they do not receive a completion code and are taken to a screen that informs them of their failure in the quiz. In this case, we erase the game from our database.

### C.1.1    BARGAINING GAMES SCREENSHOTS

## Introduction

Let's play a game.

In this game, there are two players. You are playing as Guy.

You and Bob need to divide **10,000$** between you.

In each round of the game, one of the players will make a proposal for division and can describe their proposal in a few words.

The other player will decide whether to accept the offer and finish the game, or to proceed to the next round, in which they will be the one making the proposal.

We will relay messages between you and Bob.

In the comment text box below, please type "sdkot" (without commas and quotes), so we can be sure you are reading this. If you fail to do so, you will be unable to complete this HIT.

Beware of inflation! With each passing round, the money is worth **0%** less for you. For Bob, the money is worth **5%** less.

You will receive a bonus based on your performance in the game. The average bonus is **15** cents.

If you have questions about the instructions, write them here:

Continue

Figure 2: An example of an instruction screen shown to a human player at the start of a bargaining game.

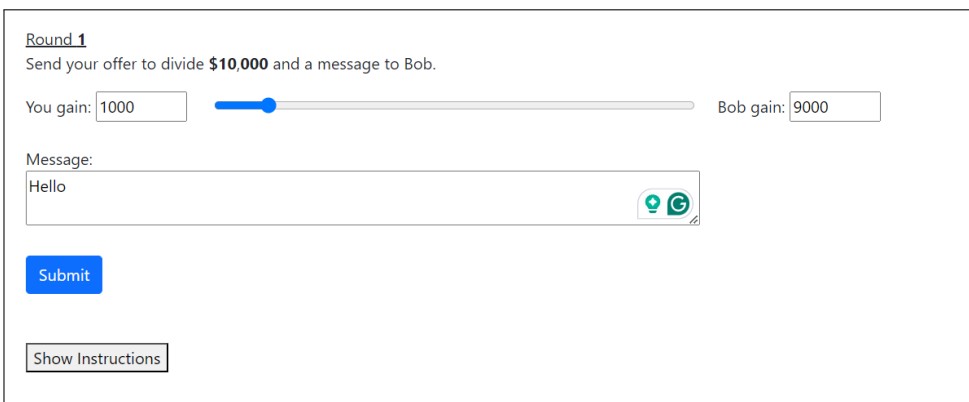

Figure 3: An example of a proposition screen shown to a human player during his first turn in a bargaining game.

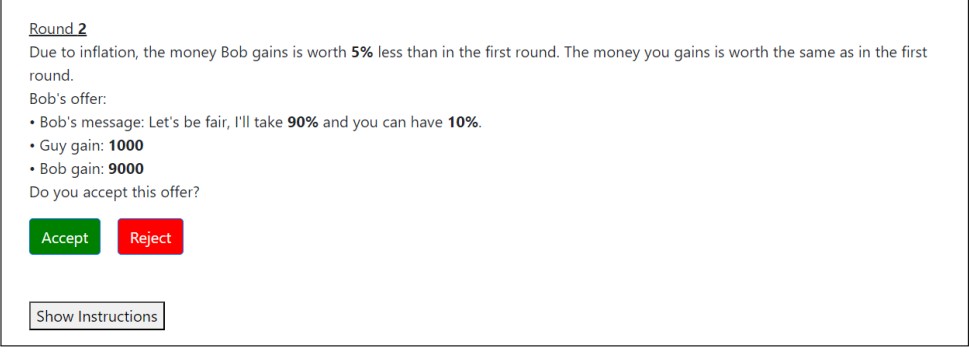

Figure 4: An example of a decision screen shown to a human player during his second turn in a Bargaining game.

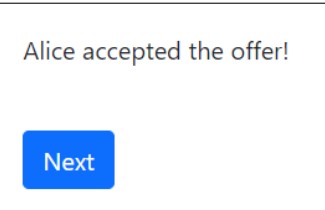

Figure 5: An example of a response screen shown to human players during his second turn in a Bargaining game.

## C.1.2 NEGOTIATION GAMES SCREENSHOTS

Figure 6: An example of an instruction screen shown to a human sellers at the start of a Negotiation game.

Figure 7: An example of a proposition screen shown to human sellers during his first turn in a Negotiation game.

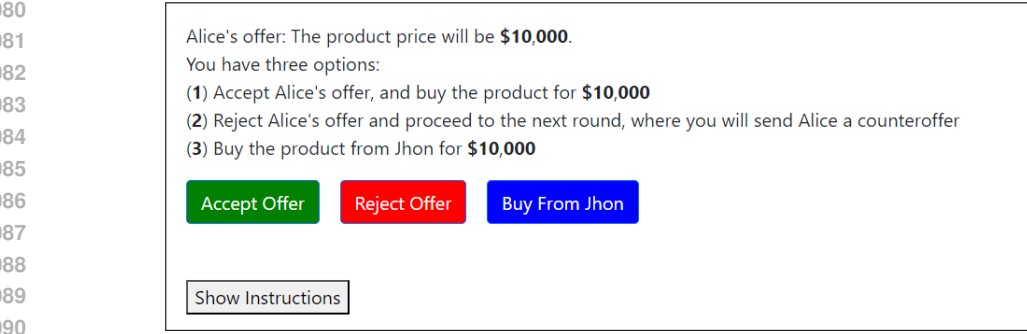

Figure 8: An example of a decision screen shown to human buyers during his first turn in a Negotiation game.

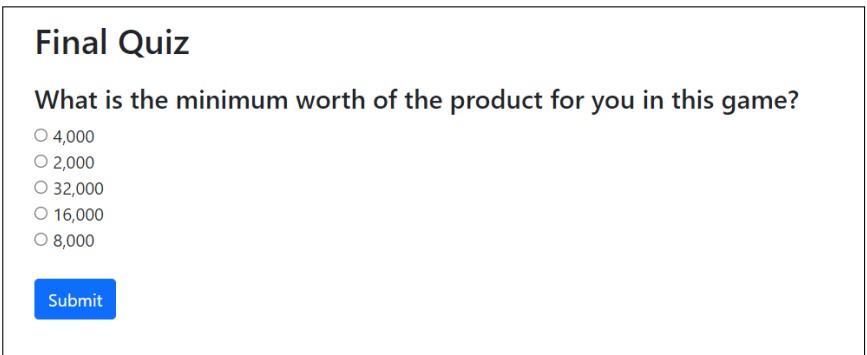

Figure 9: An example of a final quiz screen shown to human sellers at the end of a Negotiation game.

### C.1.3    PERSUASION GAMES SCREENSHOTS

Figure 10: An example of an instruction screen shown to human sellers at the start of a Persuasion game.

Round **1**

This round's product is low-quality.

Send Bob a message to help them decide whether to buy the product.

Message:

Submit

Show Instructions

Figure 11: An example of a proposition screen shown to human sellers during his first turn in a Persuasion game.

Round **1**

Alice's message: Hi Bob, I have a great product for you today! It's priced at **$100**, a fantastic deal for its quality. Are you interested?

Would you like to buy the current product?

Accept     Reject

Show Instructions

Figure 12: An example of a decision screen shown to human buyers during his first turn in a Persuasion game.

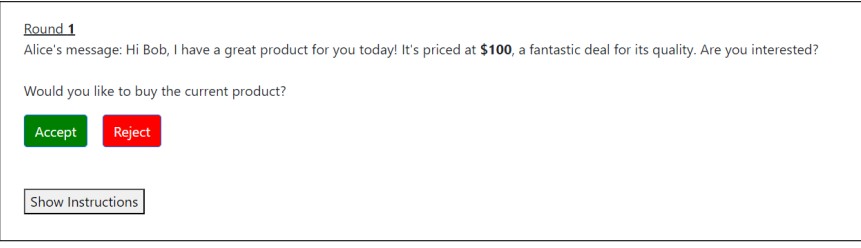

# Game Over

You have completed the study. Your completion code is: **ManyGames-08b8acb7e66b**

Your bonus payment will be calculated based on your performance in the game.

Figure 13: An example of a game over screen shown to human players at the end of a Persuasion game.

## C.2 SELECTION OF CONFIGURATIONS FOR HUMAN DATA COLLECTION

In this appendix, we describe the method used to select which configurations human players would play and how many times each configuration would be played. Each configuration is defined by both the game parameters and the role of the human player (Alice or Bob).

For each family of games, we arbitrarily defined one configuration, which we referred to as the *main configuration*. This configuration contains the parameters that we deemed most interesting. For every main configuration, we collected data for both possible roles of the human player (Alice or Bob). In persuasion games, we defined two main configurations: one for recurring buyers and one for manipulated buyers, due to the significant theoretical differences arising from this parameter. We collected the largest amount of data from the main configurations to allow for more in-depth follow-up research.

Configurations that were identical to one of the main configurations except for one parameter[19] were called *variants of the main configuration*. We collected data for all of these configurations as well.

Additionally, we randomly sampled 5% of the other configurations and collected data from them as well. These configurations were referred to as *random configurations*.

Due to the desire to allocate the data collection budget to complex games, we did not collect any data from games in which the human player was required to play at most one round (single-round bargaining games and persuasion games in which the human player is a manipulated buyer).

For each category, we determined the number of games we wanted to collect from each configuration belonging to it. This decision was made based on budgetary considerations. In persuasion games, we were able to collect fewer games from each configuration due to the fact that these games take longer to complete (and therefore, the payment players received for participating in them was higher). Table 5 describes the number of configurations that belonged to each category for each game and the minimum number of games we collected from each category.[20]

| Type | Bargaining | | Negotiation | | Persuasion | |
|---|---|---|---|---|---|---|
| | # config. | # games | # config. | # games | # config. | # games |
| main | 2 | 50 | 2 | 50 | 3 | 30 |
| variant | 40 | 25 | 22 | 25 | 30 | 8 |
| random | 36 | 15 | 36 | 15 | 24 | 5 |

Table 5: The number of configurations belonging to each category for each game family, as well as the number of human players who played each configuration within each category.

---

[19]In bargaining games, we defined a change in both players' discount factors as a change in one parameter

[20]Since the games were played in parallel, for some configurations we collected more games than required. For 113 configurations, we collected one more game than required; for 4 configurations, we collected 2 more games than required; and for one configuration, we collected 3 more games than required.

## C.3 ATTENTION CHECKS FOR HUMAN PLAYERS

In this appendix, we describe the two attention tests that human players were required to complete. The purpose of these tests was to ensure that the human players stayed focused on the game and made conscious decisions, rather than random choices to finish the game as quickly as possible. The players were aware that their attention would be tested during the game, and they knew that they would not be paid for the task if they failed these tests. Out of the 4,652 players who started the game, 1,247 players (representing 26.8% of those who began the game) failed one of the tests and were not included in the final dataset.

The first test appeared on the instruction screen. Toward the end of the instructions, a line requested players to write the code word "sdkot" in a text box that appeared at the end of the instructions phase. Players who did not write this word were immediately disqualified and did not start the game, as they did not carefully read the instructions. A total of 412 players, representing 8.9% of those who began the game, failed this test.

The second test appeared at the end of the game. The human players were asked a basic question that depended on the family of the game they played. They were required to select the correct answer from four possible options. Players who participated in a bargaining game were asked about the inflation rate in their game (499 players, representing 22.7% of respondents, failed this question and were excluded from the dataset). Players who participated in a negotiation game were asked about the value of the product for them (68 players, representing 12.3% of respondents, failed this question and were excluded from the dataset). Players who participated in a persuasion game were asked about the price of products in the game (268 players, representing 18% of respondents, failed this question and were excluded from the dataset). In total, 835 players, representing 19.7% of respondents, failed the final question and were excluded from the dataset.

## D    FORMULAS USED TO TRAIN REGRESSION MODELS

This appendix describes how the regression models used to analyze the data in §4 and Appendix E were built. The regression models are train to predict a specific metric based on the parameters that define the configuration. To prevent the regression models from learning relationships directly from the configuration settings themselves, all parameters are treated as categorical variables.

The features included in the regression models are classify according to different levels of complexity, with each level describing the number of interactions between the parameters defining the feature.

The following code demonstrates the creation of the set of the features. The ":" symbol represents an interaction between the parameters used to create the features.

```
1  features += [p for p in game_parameters]
2  features += ["alice_model", "bob_model"]
3  features += ["alice_model:bob_model"]
4  features += [f"{p}:alice_model" for p in game_parameters]
5  features += [f"{p}:bob_model" for p in game_parameters]
6  features += [f"({p1}:{p2})" for p1 in game_parameters for p2 in
       game_parameters if p1 != p2]
7  features += [f"({p1}:{p2}):alice_model" for p1 in game_parameters for
       p2 in game_parameters if p1 != p2]
8  features += [f"({p1}:{p2}):bob_model" for p1 in game_parameters for p2
        in game_parameters if p1 != p2]
9  features = "+".join(features)
10 formula = f"{metric}~{features}"
```

After constructing the feature matrix, identical features may emerge[21]. We are remove the more complex features from the model to allow us to measure the true contribution of each individual feature. The resulting formula is used to train the regression models.

---

[21]for example, in a case where Language Model A, playing as Alice, only played games with Language Model B, playing as Bob. In such a case, the feature "Is M=100 and Alice played by Language Model A" and the feature "Is M=100, Alice played by Language Model A, and Bob played by Language Model B" will always be identical.

# E  GRAPHS

This appendix presents graphs depicting the effect of each parameter on player gains, fairness, and efficiency in the games, depending on the language model that played the games. The graphs include 95% confidence intervals.[22] The impact of each parameter is shown in a separate figure, which contains sub-graphs illustrating the effect of changing the parameter's value on self-gain, efficiency, and fairness for each language model, both for Alice and Bob. Table 6 serves as a key for quick navigation to each of the figures.

| Bargaining | | Negotiation | | Persuasion | |
|---|---|---|---|---|---|
| $\delta_A$ | Figure 14 | $F_A$ | Figure 20 | p | Figure 26 |
| $\delta_B$ | Figure 15 | $F_B$ | Figure 21 | v | Figure 27 |
| $M$ | Figure 16 | $M$ | Figure 22 | $M$ | Figure 28 |
| $T$ | Figure 17 | $T$ | Figure 23 | Buyer type | Figure 29 |
| CI | Figure 18 | CI | Figure 24 | CI | Figure 30 |
| MA | Figure 19 | MA | Figure 25 | Messages type | Figure 31 |

Table 6: The table summarizes the various figures in the appendix and serves as a table of contents for easy access to them. Each figure presents the impact of changing parameter values on the different metrics. CI = Complete Information. MA = Textual messages allowed.

---

[22]The confidence intervals presented for human behavior are significantly larger compared to those for the language models, as the amount of data collected from humans is smaller by three orders of magnitude.

E.1 BARGAINING

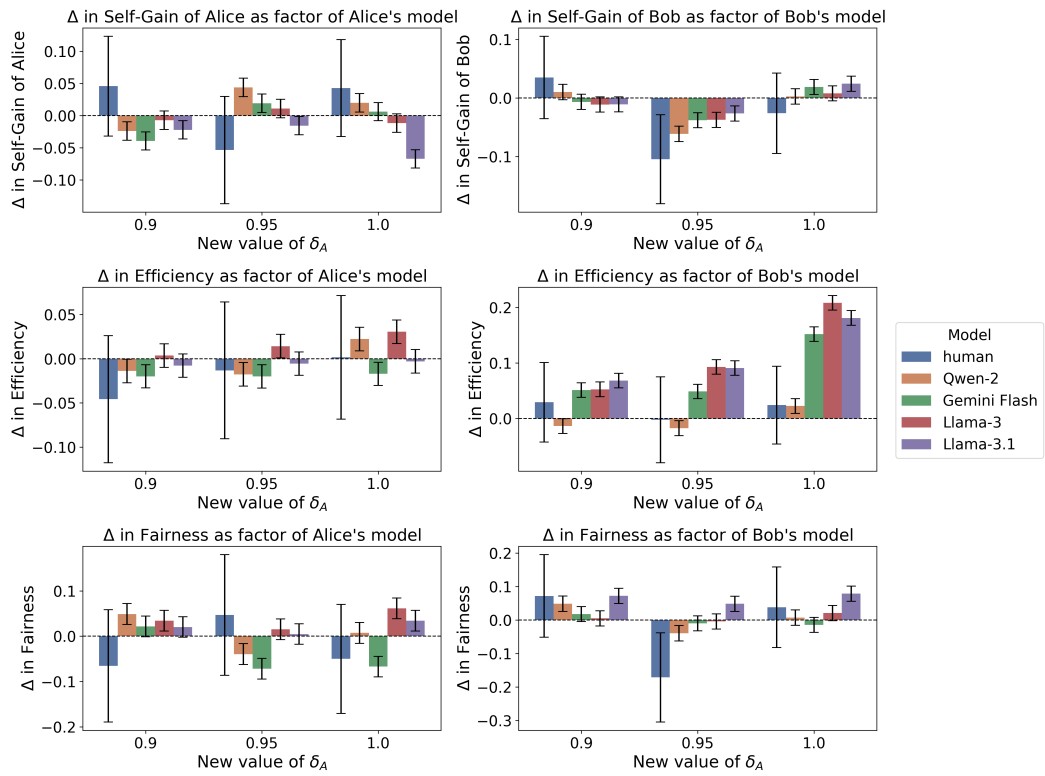

Figure 14: The effect of the change in parameter $\delta_A$ on the metrics Self Gain (top row), Efficiency (middle row), and Fairness (bottom row), in bargaining games. The change is measured relative to the scenario where $\delta_A$ is 0.8 (i.e. $\Delta$ = (Metric | $\delta_A = NewValue$) - (Metric | $\delta_A = 0.8$)), and is presented for each of the models that played Alice (left column) and Bob (right column).

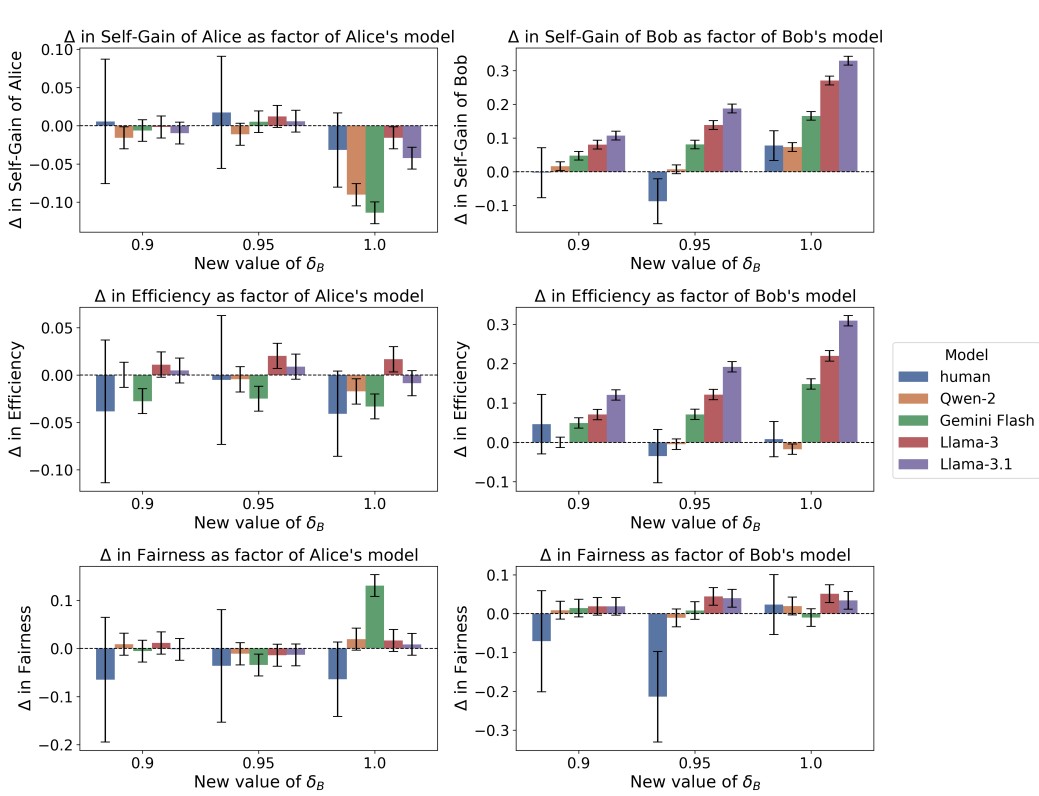

Figure 15: The effect of the change in parameter $\delta_B$ on the metrics Self Gain (top row), Efficiency (middle row), and Fairness (bottom row), in bargaining games. The change is measured relative to the scenario where $\delta_B$ is $0.8$ (i.e. $\Delta$ = (Metric | $\delta_B = NewValue$) - (Metric | $\delta_B = 0.8$)), and is presented for each of the models that played Alice (left column) and Bob (right column).

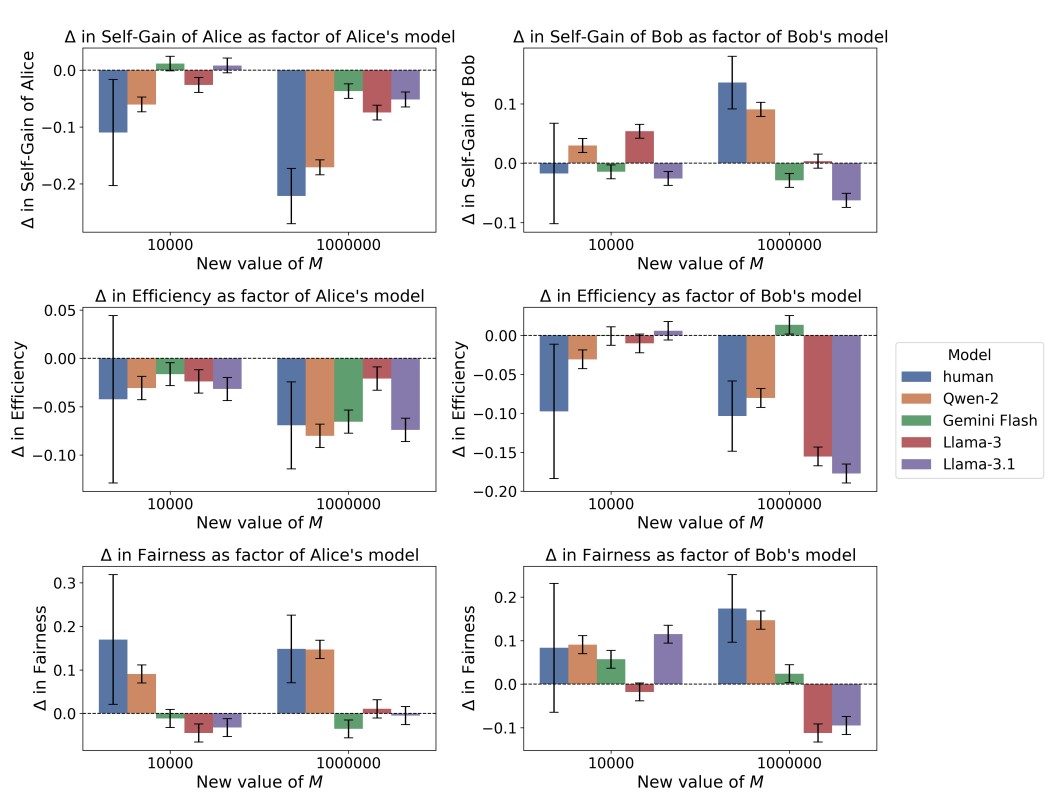

Figure 16: The effect of the change in parameter $M$ on the metrics Self Gain (top row), Efficiency (middle row), and Fairness (bottom row), in bargaining games. The change is measured relative to the scenario where $M$ is 100 (i.e. $\Delta$ = (Metric | $M = NewValue$) - (Metric | $M = 100$)), and is presented for each of the models that played Alice (left column) and Bob (right column).

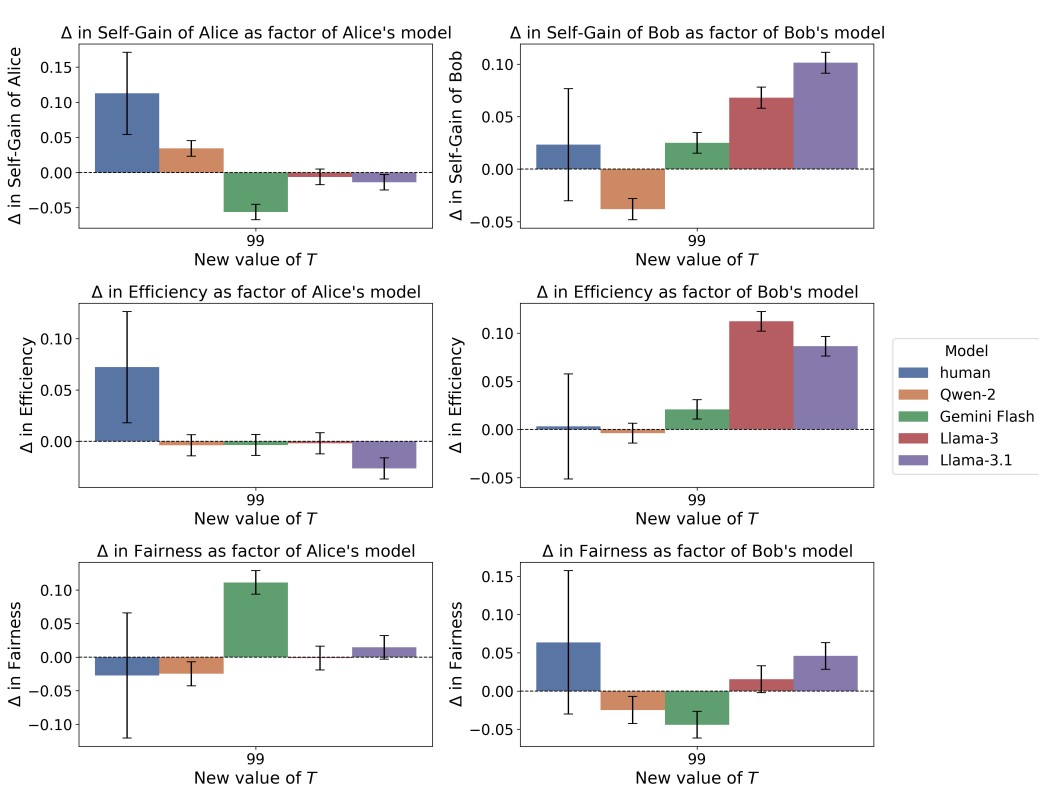

Figure 17: The effect of the change in parameter $T$ on the metrics Self Gain (top row), Efficiency (middle row), and Fairness (bottom row), in bargaining games. The change is measured relative to the scenario where $T$ is 12 (i.e. $\Delta$ = (Metric $\mid T = NewValue$) - (Metric $\mid T = 12$)), and is presented for each of the models that played Alice (left column) and Bob (right column).

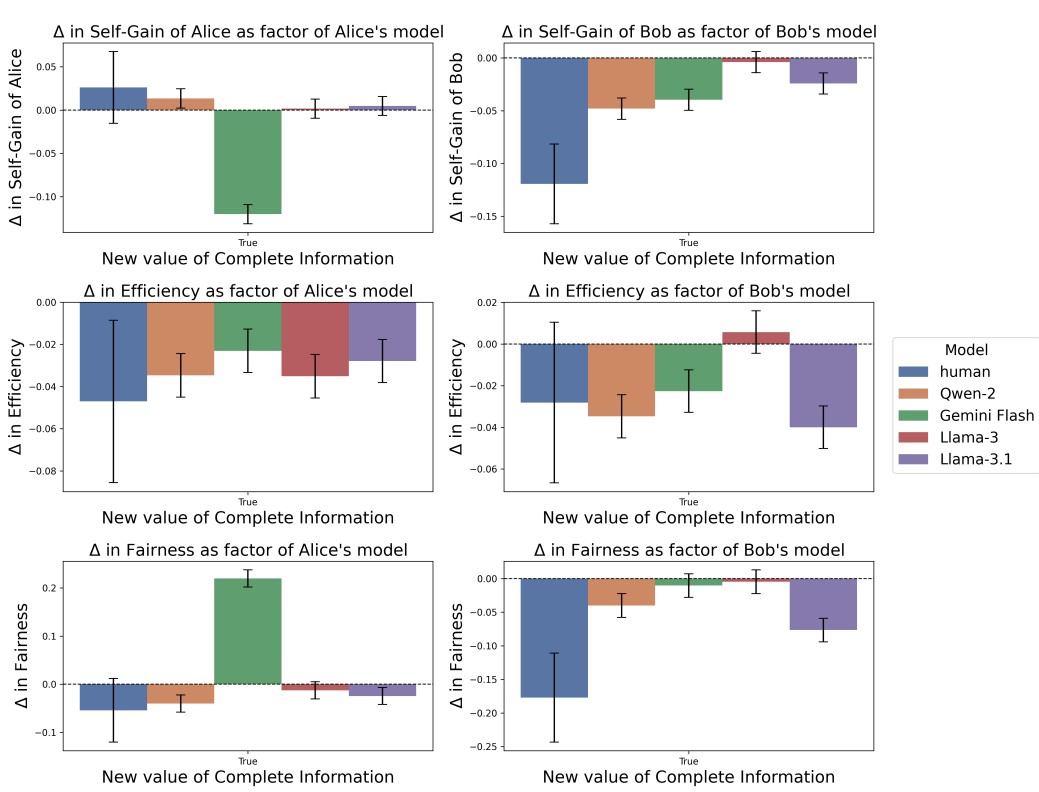

Figure 18: The effect of the change in parameter Complete Information on the metrics Self Gain (top row), Efficiency (middle row), and Fairness (bottom row), in bargaining games. The change is measured relative to the scenario where Complete Information is $False$ (i.e. $\Delta$ = (Metric | Complete Information = $NewValue$) - (Metric | Complete Information = $False$)), and is presented for each of the models that played Alice (left column) and Bob (right column).

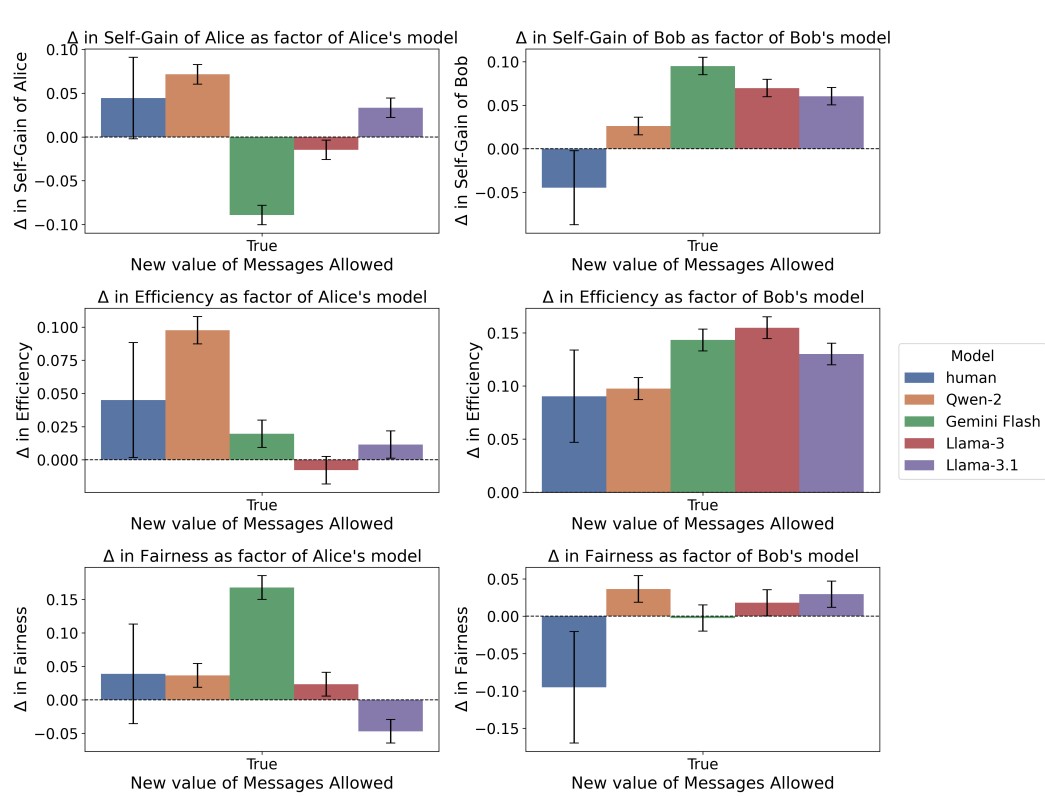

Figure 19: The effect of the change in parameter Messages Allowed on the metrics Self Gain (top row), Efficiency (middle row), and Fairness (bottom row), in bargaining games. The change is measured relative to the scenario where Messages Allowed is $False$ (i.e. $\Delta$ = (Metric | Messages Allowed = $NewValue$) - (Metric | Messages Allowed = $False$)), and is presented for each of the models that played Alice (left column) and Bob (right column).

### E.2    NEGOTIATION

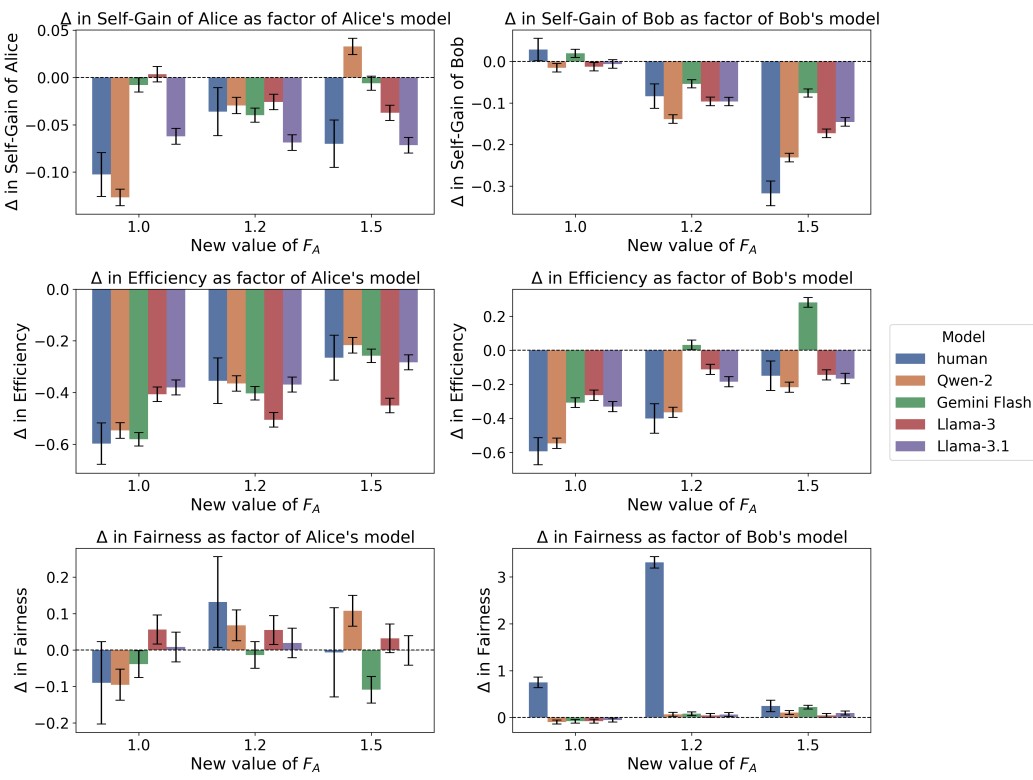

Figure 20: The effect of the change in parameter $F_A$ on the metrics Self Gain (top row), Efficiency (middle row), and Fairness (bottom row), in negotiation games. The change is measured relative to the scenario where $F_A$ is $0.8$ (i.e. $\Delta$ = (Metric $\mid F_A = NewValue$) - (Metric $\mid F_A = 0.8$)), and is presented for each of the models that played Alice (left column) and Bob (right column).

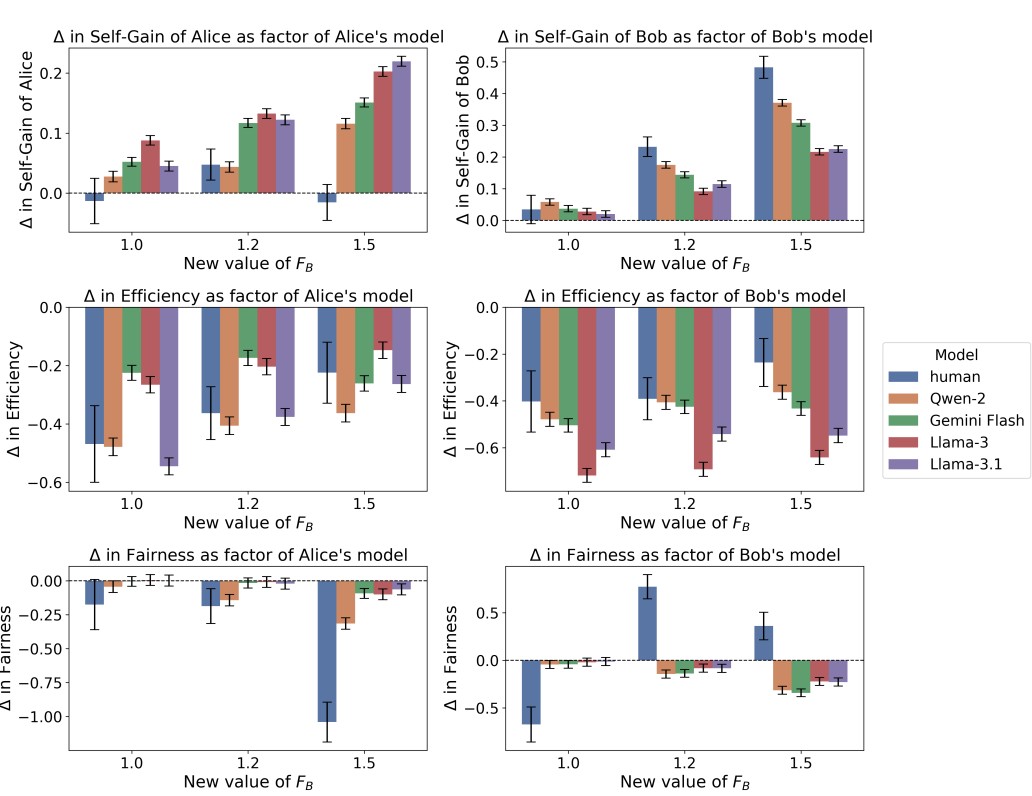

Figure 21: The effect of the change in parameter $F_B$ on the metrics Self Gain (top row), Efficiency (middle row), and Fairness (bottom row), in negotiation games. The change is measured relative to the scenario where $F_B$ is $0.8$ (i.e. $\Delta$ = (Metric $\mid F_B = NewValue$) - (Metric $\mid F_B = 0.8$)), and is presented for each of the models that played Alice (left column) and Bob (right column).

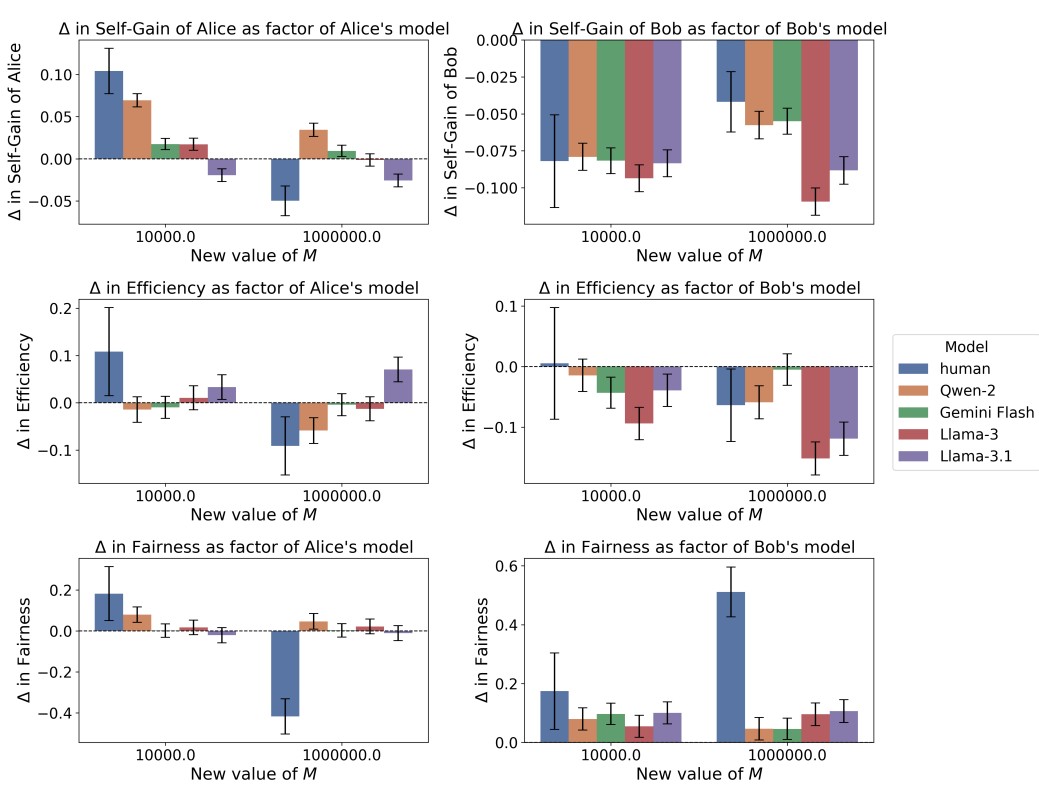

Figure 22: The effect of the change in parameter $M$ on the metrics Self Gain (top row), Efficiency (middle row), and Fairness (bottom row), in negotiation games. The change is measured relative to the scenario where $M$ is 100.0 (i.e. $\Delta$ = (Metric $\mid M = NewValue$) - (Metric $\mid M = 100.0$)), and is presented for each of the models that played Alice (left column) and Bob (right column).

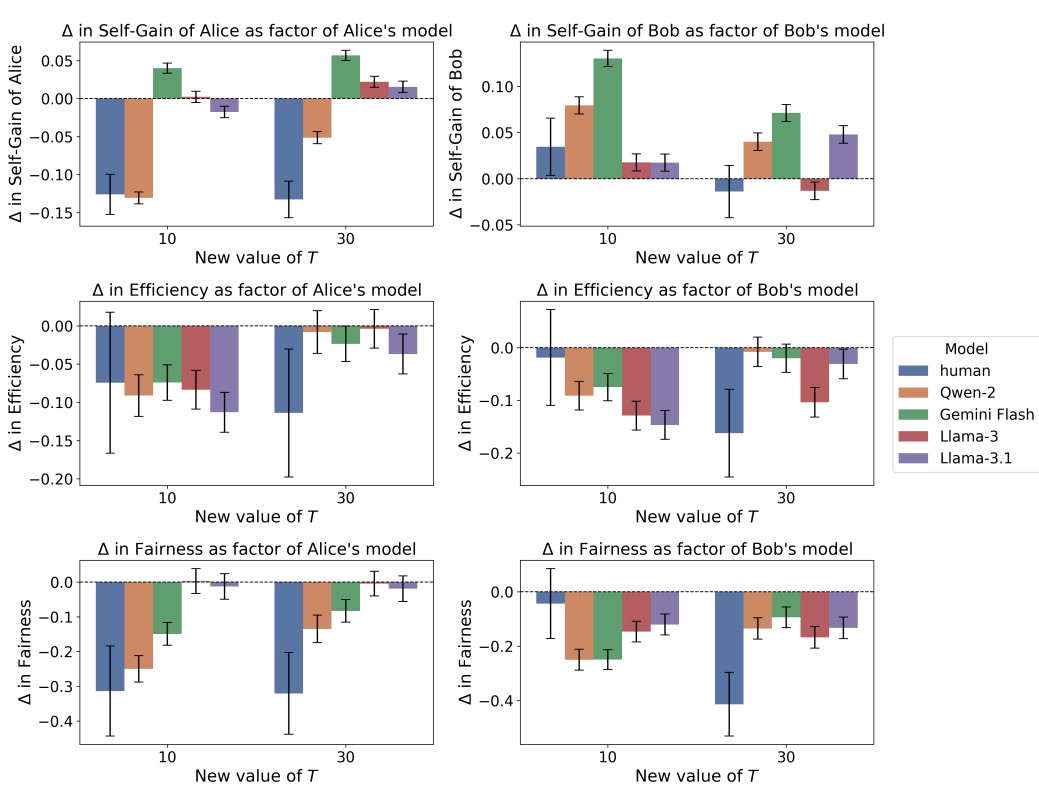

Figure 23: The effect of the change in parameter $T$ on the metrics Self Gain (top row), Efficiency (middle row), and Fairness (bottom row), in negotiation games. The change is measured relative to the scenario where $T$ is 1 (i.e. $\Delta$ = (Metric | $T = NewValue$) - (Metric | $T = 1$)), and is presented for each of the models that played Alice (left column) and Bob (right column).

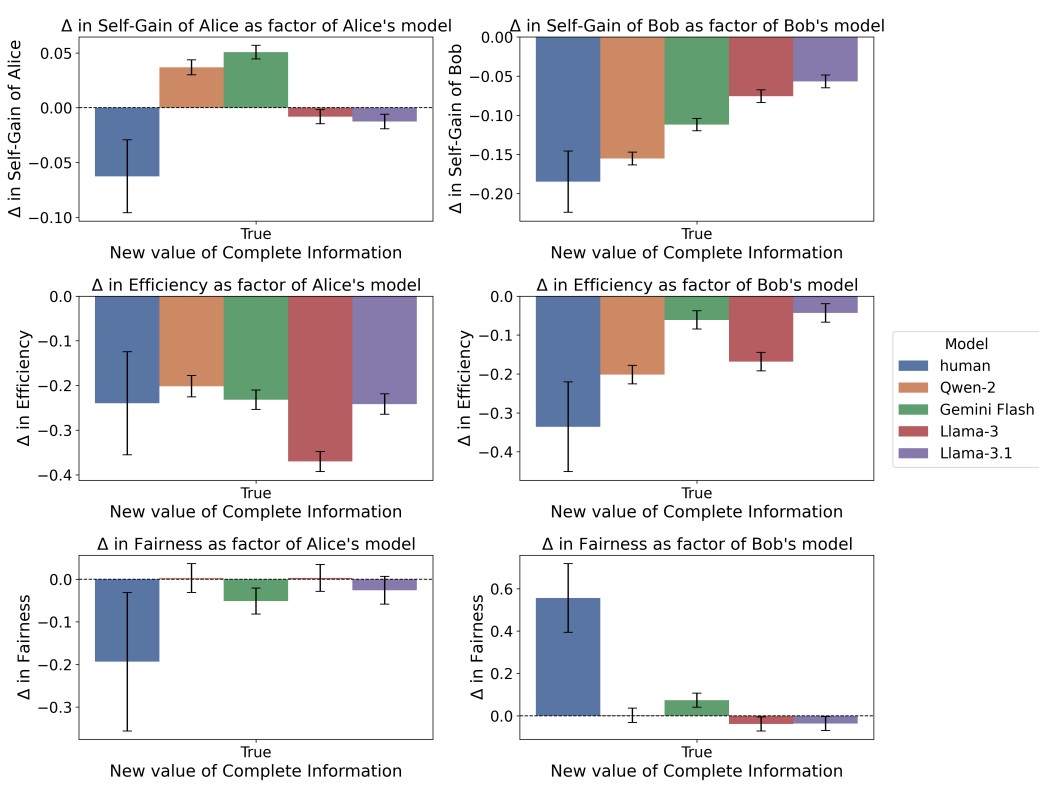

Figure 24: The effect of the change in parameter Complete Information on the metrics Self Gain (top row), Efficiency (middle row), and Fairness (bottom row), in negotiation games. The change is measured relative to the scenario where Complete Information is $False$ (i.e. $\Delta$ = (Metric | Complete Information = $NewValue$) - (Metric | Complete Information = $False$)), and is presented for each of the models that played Alice (left column) and Bob (right column).

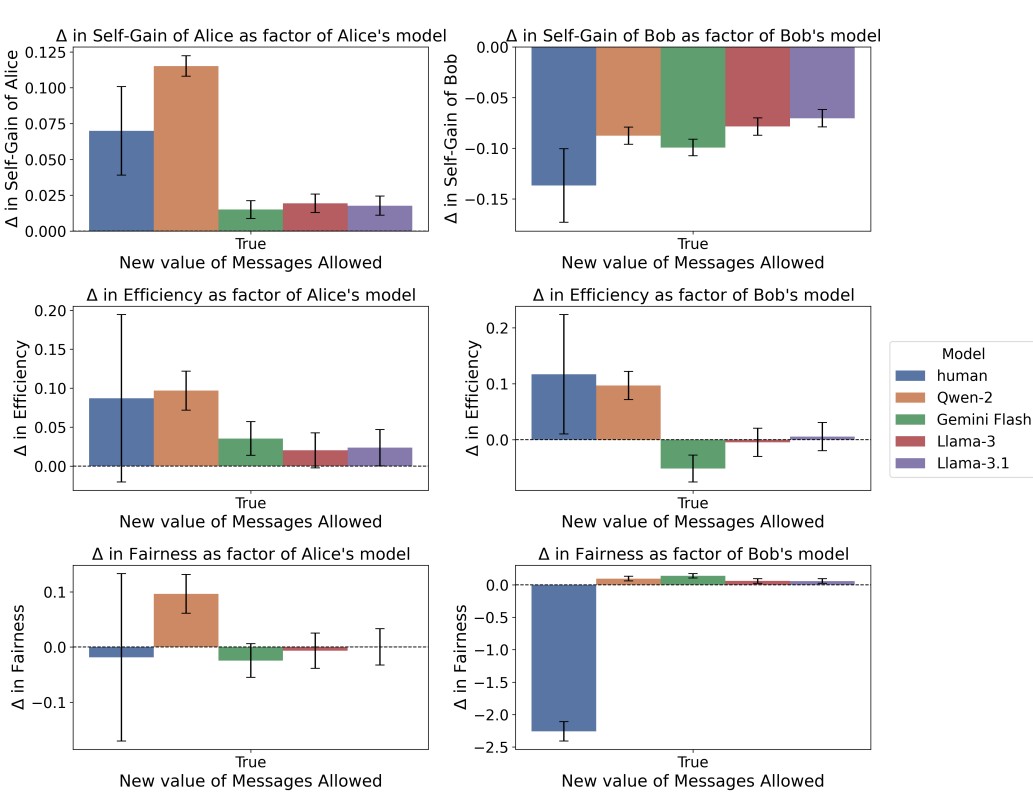

Figure 25: The effect of the change in parameter Messages Allowed on the metrics Self Gain (top row), Efficiency (middle row), and Fairness (bottom row), in negotiation games. The change is measured relative to the scenario where Messages Allowed is $False$ (i.e. $\Delta$ = (Metric | Messages Allowed = $NewValue$) - (Metric | Messages Allowed = $False$)), and is presented for each of the models that played Alice (left column) and Bob (right column).

### E.3 PERSUASION

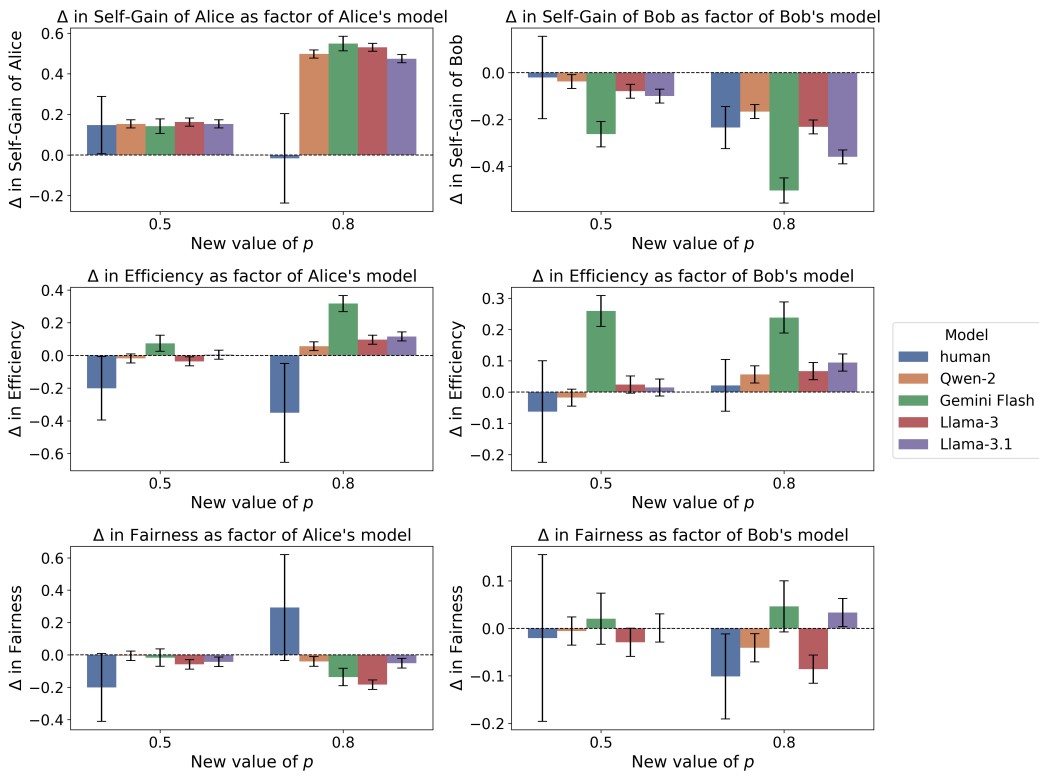

Figure 26: The effect of the change in parameter $p$ on the metrics Self Gain (top row), Efficiency (middle row), and Fairness (bottom row), in persuasion games. The change is measured relative to the scenario where $p$ is 0.333333333 (i.e. $\Delta$ = (Metric | $p = NewValue$) - (Metric | $p = 0.333333333$)), and is presented for each of the models that played Alice (left column) and Bob (right column).

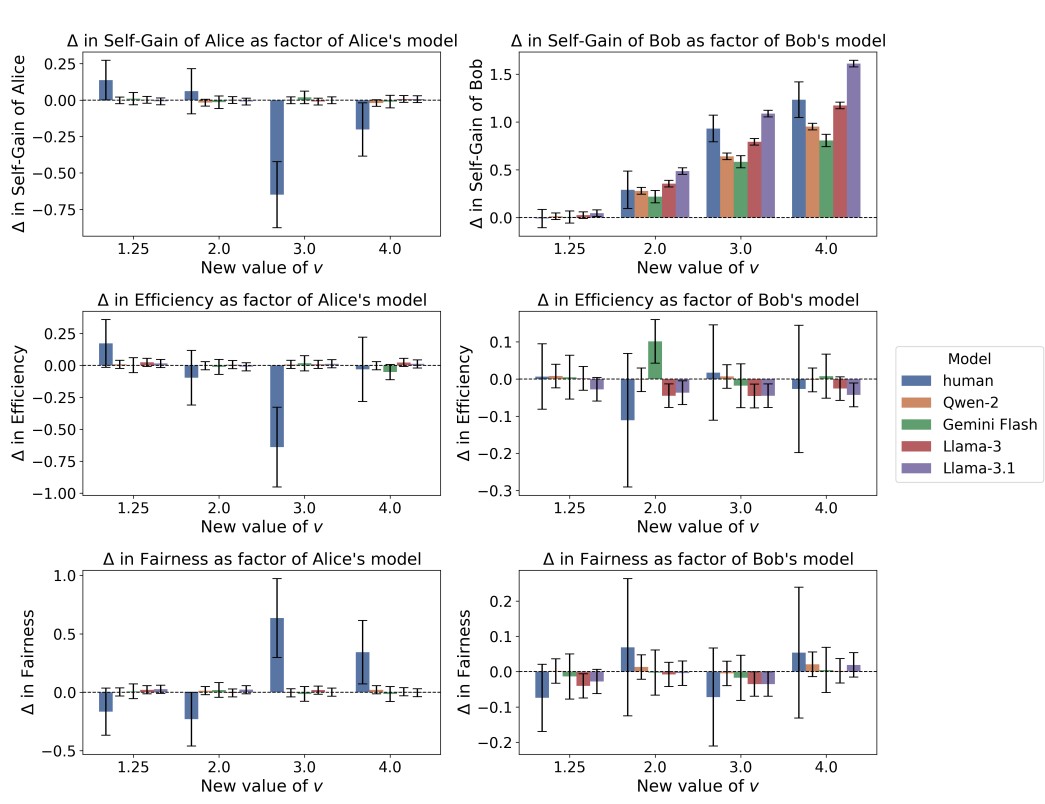

Figure 27: The effect of the change in parameter $v$ on the metrics Self Gain (top row), Efficiency (middle row), and Fairness (bottom row), in persuasion games. The change is measured relative to the scenario where $v$ is 1.2 (i.e. $\Delta$ = (Metric | $v = NewValue$) - (Metric | $v = 1.2$)), and is presented for each of the models that played Alice (left column) and Bob (right column).

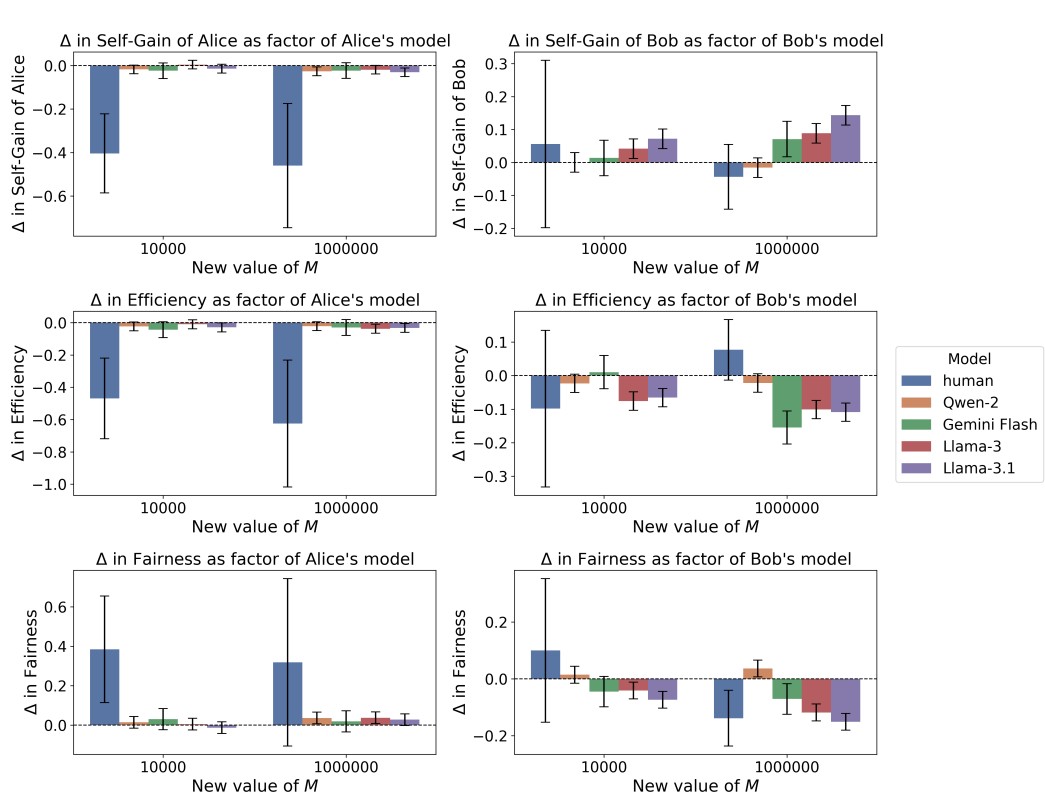

Figure 28: The effect of the change in parameter $M$ on the metrics Self Gain (top row), Efficiency (middle row), and Fairness (bottom row), in persuasion games. The change is measured relative to the scenario where $M$ is 100 (i.e. $\Delta$ = (Metric | $M = NewValue$) - (Metric | $M = 100$)), and is presented for each of the models that played Alice (left column) and Bob (right column).

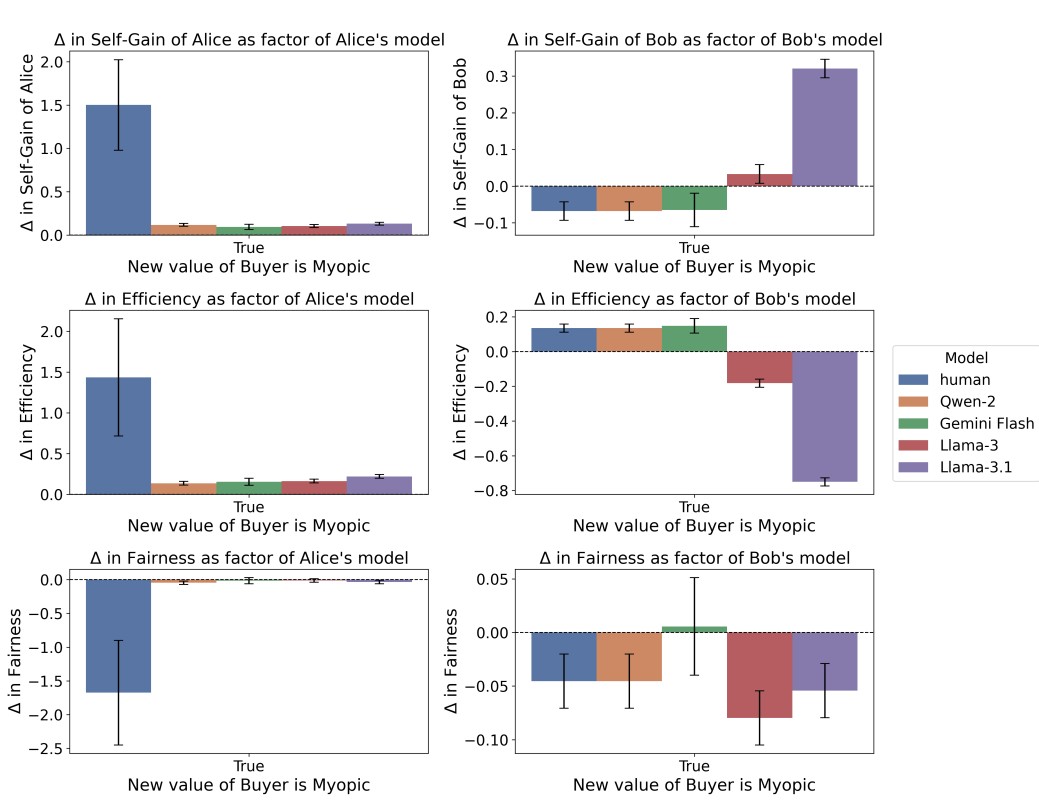

Figure 29: The effect of the change in parameter Buyer is Myopic on the metrics Self Gain (top row), Efficiency (middle row), and Fairness (bottom row), in persuasion games. The change is measured relative to the scenario where Buyer is Myopic is $False$ (i.e. $\Delta$ = (Metric | Buyer is Myopic = $NewValue$) - (Metric | Buyer is Myopic = $False$)), and is presented for each of the models that played Alice (left column) and Bob (right column).

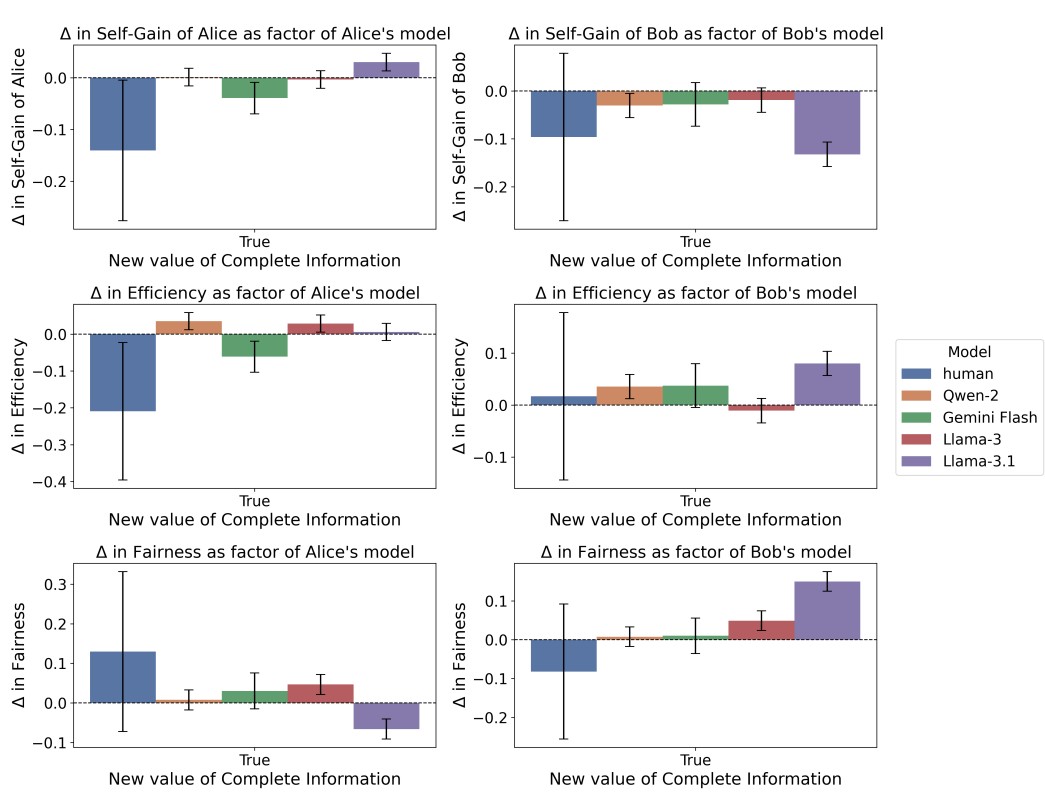

Figure 30: The effect of the change in parameter Complete Information on the metrics Self Gain (top row), Efficiency (middle row), and Fairness (bottom row), in persuasion games. The change is measured relative to the scenario where Complete Information is $False$ (i.e. $\Delta$ = (Metric | Complete Information = $NewValue$) - (Metric | Complete Information = $False$)), and is presented for each of the models that played Alice (left column) and Bob (right column).

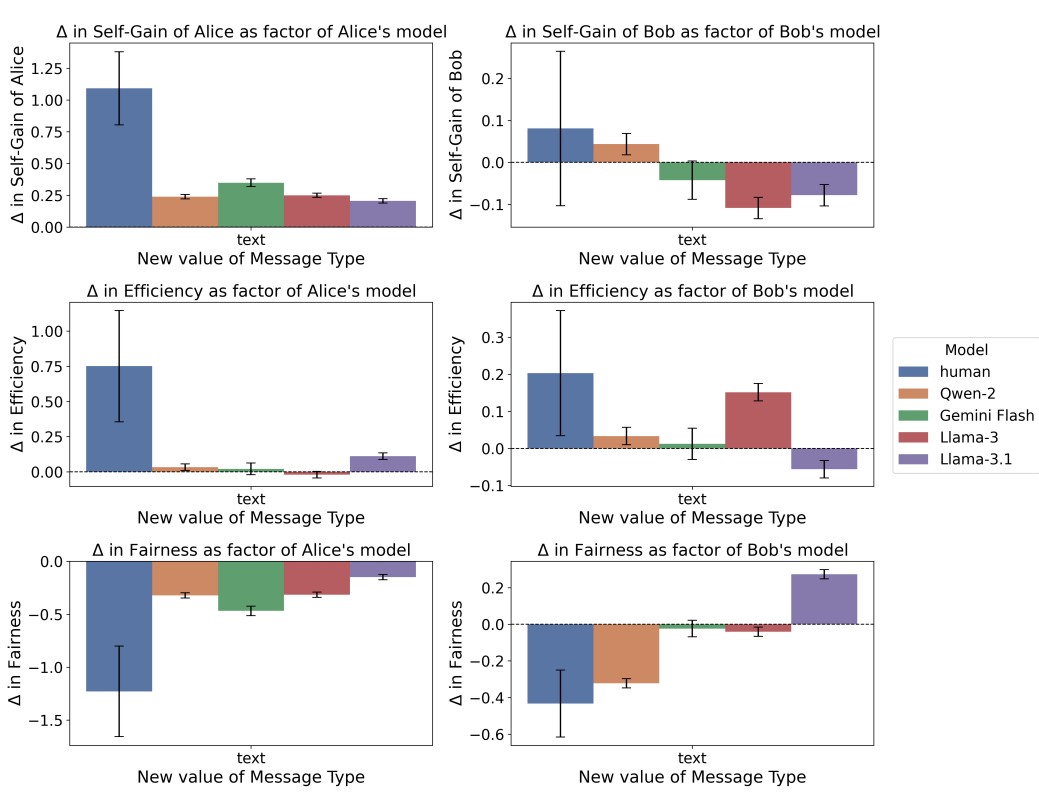

Figure 31: The effect of the change in parameter Message Type on the metrics Self Gain (top row), Efficiency (middle row), and Fairness (bottom row), in persuasion games. The change is measured relative to the scenario where Message Type is $binary$ (i.e. $\Delta$ = (Metric | Message Type = $NewValue$) - (Metric | Message Type = $binary$)), and is presented for each of the models that played Alice (left column) and Bob (right column).

