# OpenReview forum: "GLEE: A Framework and Benchmark for LLM Evaluation in Language-based Economics"
_ICLR.cc/2025/Conference — Submitted to ICLR 2025_

### Official Review · Reviewer_T6Ci · 2024-11-02

**Soundness:** 2
**Presentation:** 3
**Contribution:** 2
**Rating:** 3
**Confidence:** 4

**Summary:**

The paper introduces a framework for evaluating Large Language Models (LLMs) within economic interactions modeled as two-player sequential games. It aims to standardize research in this area by providing a comprehensive parameterization of three families of games: bargaining, negotiation, and persuasion. The framework includes a dataset of LLM vs. LLM interactions and human vs. LLM interactions, facilitating comparisons of agent behaviors and outcomes in terms of self-gain, efficiency, and fairness.

**Strengths:**

1. The paper introduces interesting classifications for agent types in economic simulations, focusing on aspects such as game horizon, information structure, and communication form.
2. The constructed simulation framework provides extensive data comparing LLMs and humans, which could be valuable for further analysis.

**Weaknesses:**

1. The framework presented in this paper lacks a bit novelty, as concepts such as bargaining and negotiation have already been addressed by existing evaluation frameworks [1, 2]. I did not find a significant distinction between this framework and prior economic frameworks.
2. Several important metrics in the analysis are not clearly defined. For example, how the economic measures of efficiency and fairness are defined and calculated in your framework?
3. The motivations of both parties are crucial components that should be integrated into the framework. For example, examining the different outcomes in bargaining scenarios, such as Alice (Fair) vs. Bob (Fair) or Alice (Selfish) vs. Bob (Fair), would provide valuable insights.

[1] How Far Are We on the Decision-Making of LLMs? Evaluating LLMs' Gaming Ability in Multi-Agent Environments
[2] Deal or no deal? end to-end learning of negotiation dialogues.

**Questions:**

See Weaknesses.

---

> ### Author Response · Authors · 2024-11-14
>
> Thank you for your review, and we are happy to address the questions and concerns raised.
>
> 1. We have addressed this concern in the response to all of the reviewers and would be happy to further discuss it. In particular, our response highlights the differences between our work and others.
>
> 2. Efficiency and fairness of all three games are defined in the last paragraphs of subsections 2: L200-L217 (bargaining), L238-248 (negotiation), and L281-287 (persuasion). Self-gain is the normalized gain of an agent in a game, and we can explicitly state it in the camera-ready if the paper is accepted.
>
> 3. Note that our framework does not inherently motivate the game players. However, this is an interesting insight, and following your review we decided to add this to our paper, i.e. providing our LLMs with some basic persona and motivation description. We will make sure to add these insights to our camera-ready version.

---

> > ### Comment · Reviewer_T6Ci · 2024-11-26
> >
> > I appreciate the authors' responses. I also think this paper would benefit much from a round of revision to integrate the suggestions from the reviews.

---

### Official Review · Reviewer_Q7Xu · 2024-11-03

**Soundness:** 3
**Presentation:** 3
**Contribution:** 3
**Rating:** 6
**Confidence:** 4

**Summary:**

The paper introduces GLEE, a framework and benchmark for assessing Large Language Models (LLMs) in economic language games, including bargaining, negotiation, and persuasion. GLEE standardizes evaluation through metrics like fairness, efficiency, and self-gain, collecting data from LLM and human interactions across multiple configurations.

**Strengths:**

The work provides a unified framework for evaluating LLMs in economic settings, which is novel and addresses the lack of standardized methods in this area.  This benchmark has various setups inspired by the economics literature covering various domains like bargaining, negotiation, and persuasion. The work also has extensive data collection, especially it collects 34k human-involved games from 3,405 players. This offers a solid foundation for analysis.

**Weaknesses:**

I in general think the work has a solid contribution to the community. However, my concerns are the games and metrics used (bargaining, negotiation, persuasion) could be limited in economic scope, potentially restricting real-world applicability. All the task setups are grounded in specific settings (e.g., Alice, Bob), which might bias the models resulting in misleading conclusions. Besides that, the work relies on only 4 LLMs without exploring a broader range of models limits the generalizability of findings.

**Questions:**

1. Could additional economic scenarios be tested to broaden applicability?
2. You define some metrics like Self Gain, Efficiency, and Fairness for different scenarios. Are there any pointers to show the validity of the definition? And would the trajectories of LLM interactions influence the scores? If not, what's the justification?

---

> ### Author Response · Authors · 2024-11-14
>
> Thank you for your review, and we are happy to address the questions and concerns raised.
>
> **Additional economic scenarios**
>
> While GLEE initially contains three families of games (bargaining, negotiation, persuasion), we highlight that one particular contribution is the source code, which is designed to be highly flexible and modular, so a researcher could easily extend its initial setups to explore various economic applications beyond our setup. In our code base, we defined an abstract “Game” class that describes the basic behavior of a game. Other researchers desiring to implement a new game simply need to create a new class that inherits from the base game class, defining their particular needs, and then will be able to use the rest of our code. Our paper, which provides parametrization, code implementation, data collection, and analysis of these games, can be seen as a proof-of-concept, that invites the community to further explore the potential (and limitations) of LLMs in language-based economics, develop new tools and techniques, etc.
>
> We also highlight that choosing these three setups as our representative games has strong justifications in the economic literature, as these three models are all fundamental, well-studied, and at the same time they have the property that unlike other models (such as auctions), real-life applications of these models often involve free language communication, as discussed in the introduction. These two properties make them natural candidates to form our framework’s base games.
>
> Finally, we will run additional models so that our findings will not be limited to 4 models.
>
> **Metrics rationale**
>
> *Self-gain* is simply the normalized utility obtained by an agent in the game, which is perhaps the most natural to measure the performance of a self-interested agent in strategic interaction. The other two metrics, efficiency and fairness, are measures of the **outcome** rather than the agent’s performance. In most games, the definition of *efficiency* (~effective utilization of productive resources) is mostly widely accepted, and in the context of our games, the original papers use these efficiency definitions for bargaining and negotiation [Rubinstein 1982; Myerson Satterthwaite 1983]. In persuasion, we view this principle as translating into “a good product should be sold” (which is a similar principle in negotiation, for example), and our metric reflects this rationale. In contrast, *fairness* is by nature highly debatable, as different notions of fairness reflect not only technical but rather distinct philosophical perspectives [Verma et al. 2018]. In our measures of fairness we focused on notions that encourage prices in the middle of seller and buyer valuations (negotiation), split that is close to equal shares among agents (bargaining), and uninformed buyers not being manipulated towards purchasing bad products (persuasion).
>
> One advantage of our framework, and in particular its technical implementation, is that it offers high flexibility in modifying existing measures and adding new ones, which can be used by researchers to explore alternative notions of fairness.
>
> **The impact of interaction trajectories on metric scores**
>
> As for the other question on trajectories, note that our measures take into account the reached outcome itself, and some of them (e.g., efficiency in bargaining games) also take into account the number of rounds it took to reach this outcome–as standard in the game theory literature. What is not directly measured using these metrics is *how* these outcomes were reached, namely, the actual behavior or communication along the gameplay. Therefore, while not all elements of the trajectories directly influence these metrics, it is clear that there is a potential *indirect effect*, such as the role of textual message exchange. This can be seen, for instance, in Figure 1: when players are allowed to exchange messages in free language, they reach a more efficient agreement (where again, efficiency measures the outcome, not the way it was achieved).

---

> > ### Comment · Reviewer_Q7Xu · 2024-11-25
> >
> > Thanks for the author's response! I would like to keep my score since I still think the setup is oversimplified (e.g., always Alice and Bob), which limits the generalization of the findings.

---

### Official Review · Reviewer_Fmqp · 2024-11-04

**Soundness:** 3
**Presentation:** 3
**Contribution:** 2
**Rating:** 5
**Confidence:** 4

**Summary:**

The paper introduces GLEE, a unified framework designed to evaluate Large Language Models (LLMs) in language-based economic games, specifically focusing on two-player, sequential interactions across three game families: bargaining, negotiation, and persuasion. The authors present a comprehensive parameterization of these games, develop an open-source framework for simulation and analysis, and provide extensive datasets from both LLM vs. LLM and human vs. LLM interactions. The work aims to standardize research in this domain, facilitating comparative studies and deeper insights into the behavior of LLM-based agents in economic contexts.

**Strengths:**

1. Comprehensive Framework: The paper presents a well-structured framework that encompasses a wide range of game configurations, inspired by established economic literature. This breadth allows for extensive exploration of LLM behaviors in diverse scenarios.

2. Data Collection Effort: The authors have invested considerable effort in data collection, amassing interactions from 954K LLM games and 3,405 human vs. LLM games. This substantial dataset is valuable for the research community.

3. Open-Source Contribution: By releasing the code and data on GitHub, the authors facilitate reproducibility and encourage further research, aligning with open science principles.

4. Integration of Human Data: Including human vs. LLM interactions provides a meaningful benchmark to assess the similarities and differences between human and artificial agents in economic decision-making.

**Weaknesses:**

1. Limited Novelty in Framework Design: While the framework is comprehensive, similar benchmarks and frameworks already exist in the multi-agent and economic game theory domains[1,2,3]. The paper does not sufficiently highlight what differentiates GLEE from existing works, nor does it clearly establish the unique advantages or novel aspects that GLEE brings to the table.

[1] Can Large Language Models Serve as Rational Players in Game Theory? A Systematic Analysis

[2] How Far Are We on the Decision-Making of LLMs? Evaluating LLMs' Gaming Ability in Multi-Agent Environments

[3] GTBench: Uncovering the Strategic Reasoning Limitations of LLMs via Game-Theoretic Evaluations

2. Limited Models: The paper utilizes models like Qwen-2-7B and Llama-3-8B, which may not represent the cutting-edge LLMs available at the time of submission, such as GPT-4o.

3. Superficial Analysis: The exploratory data analysis, while extensive, remains relatively superficial. The regression models used to predict metrics achieve moderate adjusted R-squared values (e.g., 0.57 for bargained efficiency), indicating that the models may not fully capture the complexities of the interactions. There is a lack of in-depth analysis or novel insights derived from the data.

**Questions:**

1. How does GLEE fundamentally differ from existing multi-agent and economic game theory benchmarks? Can you highlight specific features or capabilities that make GLEE uniquely valuable to the research community?

2. Beyond the parameterization and data collection, what novel theoretical insights does GLEE introduce to advance our understanding of LLM behaviors in economic settings?

---

> ### Author Response · Authors · 2024-11-14
>
> We thank you for your review and your insights, we will be glad to answer the concerns you raised:
>
> 1. We have addressed this concern in the response to all of the reviewers and would be happy to further discuss it. In particular, our response highlights the differences between our work and others.
>
> 2. We will add more models to our papers, such as GPT4-o, Llama-3.2 models, and more.
>
> 3. We also addressed this concern in the response to all reviewers, specifically while our paper provides some interesting insights into the effects of language and information on economic efficiency, our main contribution is to provide a framework for such studies.

---

> > ### Comment · Reviewer_Fmqp · 2024-11-25
> > **Thanks for the response.**
> >
> > Dear Authors,
> >
> > I have read the response.
> > I would like to keep my score since I think at least one round of revision is needed before it can be accepted.

---

### Official Review · Reviewer_CbUh · 2024-11-04

**Soundness:** 3
**Presentation:** 2
**Contribution:** 3
**Rating:** 5
**Confidence:** 3

**Summary:**

The paper proposes a benchmark to evaluate the behaviors of LLMs in economic games, with detailed parameterization on the game design and well-defined evaluation metrics. It also uses the benchmark to produce over 950k game results between LLMs, and 3405 game results involving humans for future research.  Results analysis in terms of game parameters is also provided

**Strengths:**

- The parameterized economics game design in the paper is comprehensive, and the scale of data simulation is very large.
- Using a regression model to simulate the results of human participants to compare with LLM behavior is new and interesting.

**Weaknesses:**

- The paper is not easy to read, especially when introducing the different scenarios and their degree of freedom. It would be easier if the user could use a graph to display the difference of scenarios and a table to introduce each game parameter.
- Data from humans is more inconsistent compared to LLMs due to the variance of humans; the author should give more illustration on the validity of using a regression model to model human results.
- The analysis of parameters on the final evaluations is not enough; it is hard to find clear conclusions from the main part of the paper.Also, the figures, for example, in Figure 1, are hard to comprehend.

**Questions:**

- In line 447-449, the author mentions Human achieves the worst performance when playing Bob and poor performance in negotiation game, which is counterintuitive. Can the author give more analysis on the results?
- In Lines 499-500, Could the decrease of efficiency under the full information originates from the model's limited capability of dealing with long context?

---

> ### Author Response · Authors · 2024-11-15
>
> We thank you for your review and are happy to address the concerns you raised:
> 1. We will work on making the paper easier to read, specifically the game introduction part.
> 2. Other papers found that humans have much more variation in their answers than LLMs. One explanation states that, after having pretrained on the entire internet, LLMs can be considered as the average of many humans and therefore show less variation. However, despite this difference in consistency vs. inconsistency, regressions have been used to model human responses with some success [1].
> 3. We hope we have addressed this concern in the general answer to reviewers.
> 4. The phenomenon you mentioned is indeed an interesting one. After examining the game data, it appears that the phenomenon arises from the tendency of LLMs to make relatively low opening offers (an average of 0.21 of the total sum), whereas humans make more fair offers (an average of 0.39 of the total sum). Nevertheless, it is also evident that humans and LLMs agree on the first offer at a similar rate. Humans' acceptance of lower offers may stem from the time-loss element. Humans have a hidden interest in finishing the game as quickly as possible, while LLMs are more aggressive, making ambitious offers and taking the risk that the negotiation may extend, as long as it increases their chances of securing a larger profit in the end.
> 5. Note that the difference between a full information setup and a partial information one is in the basic prompt with just a few characters of difference (for example, whether the exact price of the product is mentioned or not). However, the free language setup allows to send full and long messages to the other player, whereas the structured output setup sends concise and precise messages. We saw in our paper that the language setup leads LLMs to an increase in efficiency, hinting that the length of the context does not impair LLM performance in our setup.
> [1] Samuel Amouyal, Aya Meltzer-Asscher, and Jonathan Berant. 2024. Large Language Models for Psycholinguistic Plausibility Pretesting. In Findings of the Association for Computational Linguistics: EACL 2024

---

> > ### Comment · Reviewer_CbUh · 2024-11-28
> >
> > Thank the authors' efforts on the response. I think it would be better for the author to integrate these content in the revised version to give a clearer explanation in the context.

---

### Author Response · Authors · 2024-11-14
**General Authors Response**

Thank you for your thoughtful and constructive feedback, which we believe will greatly enhance the quality of our paper. We would like to address key points regarding the contribution and novelty of our work, as several reviewers raised questions on these aspects.

Our work is the first to offer a parametrization of two-player, sequential, **language-based** games. In these games, players can exchange natural language messages, expanding beyond the stylized, predefined messages often examined in economic literature. These games are at the core of information economics.

Some of the reviewers mentioned works that establish valuable game-theoretic benchmarks for LLM performance. However, these benchmarks largely treat language as a technical tool rather than an integral part of the strategy. For instance, in frameworks where LLMs play repeated prisoner’s dilemma or rock-paper-scissors, language is only used to announce or observe moves, with the strategy space confined to defect/cooperate or rock/paper/scissors. This approach contrasts with our framework, where the content of each message is a critical element of the strategy itself, as discussed in our introduction (and also demonstrated empirically). This is the case in [1,2,3], but it is indeed worth mentioning that these have their advantages, such as studying novel aspects such as beliefs evolvement, or other foundational games such as sealed bid auctions (whose practical applications typically do not involve language communication).

While other studies have examined LLMs in economic settings involving language—such as [4] on negotiation and [5] on persuasion—these studies are limited to specific game types, making it challenging to generalize their findings. Our work, by contrast, introduces a consistent parametrization that enables comparisons **across game families**, allowing researchers to explore questions like, “What are the effects of language, information, or game horizon on self-gain, efficiency, or fairness in persuasion games versus negotiation games?” This flexibility broadens the scope of insights achievable through behavioral economics, which often limits itself to overly simplified, context-specific interactions.

In fact, our parametrization is novel even **within game families**, as most of the previous work has focused on setups that are relatively specific and application-oriented, such as [5] that considered repeated persuasion in the context of hotels and travel agents, in which agents always have complete information about each others’ preferences (i.e., not allowing for the ‘incomplete information’ in our terminology).
In addition, our open-source codebase is designed to be high-quality, modular, and flexible, allowing researchers to extend it by adding new games or evaluation criteria, such as alternative notions of fairness tailored to specific applications.
Together, these contributions represent a meaningful advancement toward LLM-based experimental economics, providing researchers with a scalable framework to study diverse economic mechanisms.

As some reviewers noted, the scope of our analysis and results is somewhat limited. While our analysis does provide interesting insights, such as the effect of information and language on economic efficiency (Section 4, Q3), we wish to highlight that we view this mostly as an example of the potential in our framework. That is, the primary focus of our work is on the (a) parametrization of the (fundamental, language-based) game space; (b) release of code and data; and, most importantly, (c) raising new research directions that necessitate interdisciplinary collaboration. For example, our work intersects NLP (for language interactions), theoretical and behavioral economics (for complex economic models), and machine learning (for pattern recognition in high-volume, high-variance data).

We find these potential collaborations as possessing true scientific value, and the purpose of GLEE is to make it more accessible (technically and conceptually) to researchers from these domains. We are very excited about these opportunities and hope that the community will accept this call to arms.

Finally, if the paper is accepted, we would be glad to emphasize these points in the camera-ready version of our paper.



**References:**

[1] Can Large Language Models Serve as Rational Players in Game Theory? A Systematic Analysis

[2] How Far Are We on the Decision-Making of LLMs? Evaluating LLMs' Gaming Ability in Multi-Agent Environments

[3] GTBench: Uncovering the Strategic Reasoning Limitations of LLMs via Game-Theoretic Evaluations

[4] Deal or no deal? end-to-end learning of negotiation dialogues

[5] Can LLMs Replace Economic Choice Prediction Labs? The Case of Language-based Persuasion Games

---

### Meta-Review · Area_Chair_yq1L · 2024-12-08

**Metareview:**

The paper introduces GLEE, a unified framework to evaluate LLMs in language-based economic games, focusing on two-player sequential interactions across bargaining, negotiation, and persuasion scenarios. It provides an extensive parameterization of game designs, a large dataset of LLM vs. LLM and human vs. LLM interactions, and an open-source implementation for reproducibility.

While the paper provides a valuable dataset and a comprehensive framework for studying LLM behavior in economic games, it lacks novelty compared to existing works, relies on a limited set of models, and offers only superficial analysis. Improvements in clarity, model diversity, metric definitions, and deeper insights into the results are necessary for the paper to make a significant impact. All reviewers agree that significant improvement is needed for acceptance. Therefore, I recommend rejection at this time.

**Additional Comments On Reviewer Discussion:**

Most of the reviewers think the paper's contribution is limited, lack of in-depth analyses and discussion, and paper writing could be largely improved for better understanding. Given these require great efforts and cannot be done during rebuttal, no changes were made after the authors' response.

---

### Decision · Program_Chairs · 2025-01-22

Reject